# Structural and biochemical basis for activity of *Aspergillus nidulans* α-1,3-glucanases from glycoside hydrolase family 71

Scott Mazurkewich [1,2] ✉, Tove Widén [1], Hampus Karlsson[2,3,4], Lars Evenäs [2,3,4], Poornima Ramamohan[2,5], Jakob Wohlert[2,5], Gisela Brändén [6] & Johan Larsbrink [1,2] ✉

The microbial polysaccharide α-1,3-glucan is an important component of fungal cell walls and dental plaque biofilms, contributing to microbial virulence and biofilm resilience. Glycoside hydrolase family 71 (GH71) includes α-1,3-glucan degrading enzymes which could be exploited for biotechnological applications; however, the family is presently poorly understood. To increase our understanding of GH71, we have performed a phylogenetic analysis of the family and detailed biochemical analysis of two of the five GH71 enzymes encoded by *Aspergillus nidulans* (*An*GH71B and -C). Both are active on soluble α-1,3-glucooligosaccharides but surprisingly only minimally on water-insoluble α-1,3-glucan. Assays on intact and milled *A. nidulans* biomass indicate that the enzymes act on fungal cell wall glycosidic linkages, likely having roles in cell wall remodelling. Both enzymes utilize an inverting mechanism but differ in specificity and product profiles indicating *exo*- and *endo*-like activity for *An*GH71B and *An*GH71C, respectively. We present the first structure of a GH71 protein, *An*GH71C, including structures with carbohydrate ligands. These structures revealed a conserved acidic dyad (DxxE), found to be crucial for activity, and active site water coordination consistent with a classical inverting GH mechanism. This work provides new insights into GH71, highlighting its functional diversity and the enzymes roles in fungal physiology.

α-1,3-glucan is a water-insoluble polysaccharide present in the cell wall of many fungal species, including filamentous fungi (e.g. *Aspergillus* species, *Magnaporthe grisea*), dimorphic fungi (e.g. *Histoplasma capsulatum*, *Paracoccidioides brasiliensis, Blastomyces dermatitidis*) and in yeasts (e.g. *Schizosaccharomyces pombe*)[1]. Many of these fungi are pathogenic, mainly affecting immunocompromised individuals. Invasive fungal infections have high mortality rates, 20–95%, with *Aspergillus fumigatus* alone estimated in 2012 to be responsible for over 200,000 life-threatening infections annually world-wide[2,3]. In some of these pathogenic species, α-1,3-glucan has been shown to act as a virulence factor through various mechanisms, in addition to being a cell wall structural component. For example, in the plant pathogen *M. grisea* and the human pathogen *H. capsulatum*, α-1,3-glucan masks the β-glucans in the cell wall thus protecting it from host recognition and immune responses[4,5]. In the pathogenic yeast *C. neoformans*, the causative agent of meningitis in humans, α-1,3-glucan anchors capsular polysaccharides to the cell wall and mutants lacking α-1,3-glucan production have reduced or modulated virulence[6,7]. The role of α-1,3-glucan in *A. fumigatus* is somewhat unclear, though it is important for the viability of conidia, and deletion of genes controlling its biosynthesis leads to reduced virulence[8]. The polysaccharide is also found in the exopolysaccharide matrix of the dental plaque biofilm where it is produced by bacteria such as *Streptococcus mutans*, from which the eponymous name mutan for the glucan is derived. It is one of the main polysaccharides promoting dental plaque build-up, enabling bacteria to cluster together and attach to the tooth

[1]Department of Life Sciences, Chalmers University of Technology, Gothenburg, Sweden. [2]Wallenberg Wood Science Center, Chalmers University of Technology, Gothenburg, Sweden. [3]Department of Chemistry and Chemical Engineering, Chalmers University of Technology, Gothenburg, Sweden. [4]FibRe-Centre for Lignocellulose-based Thermoplastics, Chalmers University of Technology, Gothenburg, Sweden. [5]Department of Fibre and Polymer Technology, KTH Royal Institute of Technology, Stockholm, Sweden. [6]Department of Chemistry and Molecular Biology, University of Gothenburg, Gothenburg, Sweden. ✉e-mail: scott.mazurkewich@chalmers.se; johan.larsbrink@chalmers.se

enamel[9]. If left untreated, dental plaque can lead to more severe oral diseases such as caries, gingivitis and periodontitis that cause substantial pain and economic strain on individuals and societies[10,11]. The polysaccharide can be degraded by enzymes known as either mutanases or α-1,3-glucanases but, despite its vital role in fungal virulence and dental biofilms, little is known about its enzymatic degradation.

Two enzyme families in the carbohydrate active enzymes database (CAZy, www.cazy.org) contain enzymes with α-1,3-glucanase activity: glycoside hydrolase families 71 and 87 (GH71 and GH87[12]). Although our fundamental understanding of these enzymes is limited, some potential applications have been explored such as degradation of α-1,3-glucan-containing biofilms in vitro[13], reducing dental plaque in vivo[14,15], enhancement (or potentiation) of the effect of antimicrobial agents towards cariogenic bacteria[16], and there is also a report linking improved barley straw saccharification by enzyme cocktails to presence of GH71 enzymes[17], though the increase in released sugars could stem from other factors than direct straw hydrolysis, as α-1,3-glucan is not a known component of barley straw or other cereals. Further, rice expressing an α-1,3-glucanase gene showed resistance towards several fungal pathogens, including *Magnaporthe oryzae*, *Cochlioborus miyabeanus* and *Rhizoctonia solani*[18]. Of the two families, GH71 is particularly interesting as few members have been biochemically characterized and no protein structural information was present, precluding a deeper understanding of the function the enzymes. Bioinformatic analysis of GH71 as a whole is limited; earlier investigations has suggested the fungal portion of GH71 can be categorized into 5 groupings[19], while little has yet been done to compare with bacterial members. Several fungal organisms encode multiple enzymes distributed across the fungal portion of the phylogenetic tree (see Results section). One such case is the filamentous fungus *Aspergillus nidulans*, which intriguingly encodes five putative GH71 enzymes. Only one of these (MutA, herein termed as *An*GH71C) has been

partially characterized and confirmed to be active on α-1,3-glucan[20,21]. Expression studies of the genes encoding *An*GH71C, AgnE (herein termed *An*GH71A), and AgnB (herein termed *An*GH71B), have revealed differential expression depending on the fungal life cycle stage, and overexpression of *An*GH71A and -B led to a decreased cellular α-glucan content[20,22].

Here, we significantly expand the fundamental knowledge of GH71 and its fungal subfamily through detailed biochemical characterization of *An*GH71B and *An*GH71C, and provide the first crystal structure of an enzyme in the family. We show that, while specifically cleaving α-1,3-linkages, the enzymes preferentially act on oligosaccharides rather than large α-1,3-glucan polysaccharides, and they differ in their specificity for oligosaccharide chain length. The structure of *An*GH71C was determined by macromolecular crystallography and structures of ligand complexes with oligosaccharides revealed an active site with at least seven subsites for oligo- or polysaccharide recognition. Together the results shed light on a relatively unexplored glycoside hydrolase family whose enzymes could be exploited for various biotechnological applications.

## Results
### Bioinformatic analyses

To deepen the understanding of GH71 and to provide a basis for structure-function studies, we examined the phylogeny of all annotated members present in the CAZy database. At present, GH71 consists principally of bacterial and fungal enzymes, and the tree is split with a clear separation between the sequences from the different kingdoms, which we here refer to as subfamilies (Fig. 1). Currently there are only two plant GH71 entries in CAZy, both from Sitka spruce (*Picea sitchensis*, GenBank: ACN40311.1 and ACN40737.1), but a broader sequence search with BLAST[23] using the GH71 entries from Sitka spruce as queries revealed homologs in other plant species, such as Chinese yew (*Taxus chinensis*, KAH9289368.1), Japanese cedar (*Cryptomeria japonica*, XP_057847126.1), and some mosses and liverworts, indicating that the presence of GH71 enzymes in plantae might be underappreciated. Modularity analysis revealed that about half of the GH71 members are multidomain proteins, with additional domains such as carbohydrate binding modules (CBMs) and, more rarely, additional predicted catalytic domains, such as domains from GH18, glycosyl transferase family 4 (GT4), and a conserved domain of unknown function (DUF) (Supplementary Fig. 1). CBMs from families 24 and 96 are the most commonly observed, with α-1,3-glucan binding found in CBM24[24] and alginate binding found in CBM96[24], though few members from either family have been characterized to date. Interestingly, although the alignment used to create the core phylogenetic tree was carried out with sequences trimmed to contain only the GH71 domain, members with additional domains typically clustered together, suggesting similar evolutionary origins of such architectures. Fungal species frequently encode multiple GH71 members, often between four and ten homologs, while bacterial species instead often encode only one or two members (Fig. 1). The five *A. nidulans* GH71 members (*An*GH71A through -E, named based on the order of their respective Genbank accessions: AN1604.2, AN3790.2, AN7349.2, AN8252.2, and AN9042.2) collectively share sequence identities ranging from 31 to 45% and are thus distinct. The *A. nidulans* enzymes are furthermore distributed throughout the fungal subfamily, which makes the species a promising model for understanding GH71 diversity.

### Enzyme characterization reveals distinct specificities for nigerooligosaccharides

Two of the *A. nidulans* GH71 members – *An*GH71B and *An*GH71C – were successfully produced through recombinant expression in *E. coli* and subsequent purification by immobilized affinity chromatography to yields > 100 mg/L culture. However, attempts to recombinantly produce *An*GH71A, *An*GH71D and *An*GH71E were unsuccessful despite extensive efforts in *E. coli* and *P. pastoris* (see Materials and Methods). Screening assays with a range of polysaccharides were used to assess the activity profiles of *An*GH71B and *An*GH71C; both enzymes cleaved α-1,3-glucan

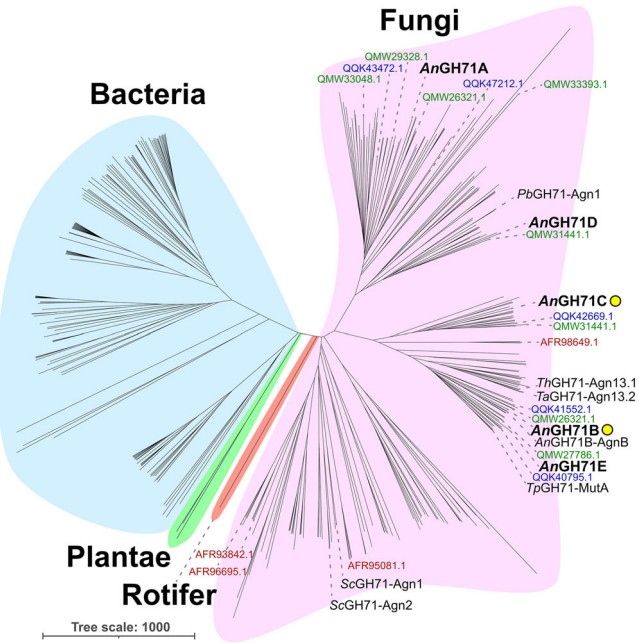

**Fig. 1 | Phylogenetic tree of GH71.** The unrooted tree of GH71 was created with iTOL[63] using curated sequences (355) from CAZy as described in the methods. The family is currently principally composed of fungal and bacterial members, coloured in fuchsia and blue, with members found in the plant Sitka spruce (*Picea sitchensis*, green) and the rotifer *Adineta vaga* (red). Characterized enzymes are indicated with their name and respective Genbank identifier and members from *A. nidulans* FGSC A4 are identified in bold text with *An*GH71B and -C additionally identified by a yellow circle. The Genbank identifiers of GH71 genes from *Cryptococcus neoformans* var. grubii H99 (red), *Penicillium digitatum* PdW03 (blue), and *Aspergillus flavus* NRRL3357 (green) are shown as an example of fungal species coding for multiple GH71 proteins.

and completely lacked activity on any other tested α-linked polysaccharides (dextran, starch, α-mannan from *S. cerevisiae*, isolichenan, and alternan, the latter two being mixed-linkage α-glucans including α-1,3 linkages) or on any β-linked polysaccharides (barley β-glucan, yeast β-glucan, curdlan, konjac glucomannan, ivory nut mannan, micro crystalline cellulose, or carboxymethyl cellulose). In contrast to the recently characterized α-1,3-glucanase *Fj*GH87[25], only a small proportion of the α-1,3-glucan used in the assays (<1%, determined by reducing sugar assay with 3,5-dinitrosalicylic acid) was hydrolysed by the GH71 enzymes over a 20-h period (Fig. 2a, c), and the amount of glucose and small oligosaccharides produced only minimally increased after 120 min (Fig. 2b, d). Interestingly, the products produced by the enzymes differed, with *An*GH71B principally producing glucose and a smaller amount of nigerotriose, while *An*GH71C instead released a spread of products, from glucose to nigerotetraose. Notably, both enzymes appeared to favourably target the large and presumably semi-soluble oligo- or polysaccharides present in the substrate preparation (insets, Fig. 2b, d), which could indicate that these are the preferred substrates rather than large molecular weight insoluble polysaccharide, which would also explain the slowing reaction progress observed once they had been depleted.

To explore the enzymes' oligosaccharide utilization, we characterized their specificity with commercially available nigerooligosaccharides, ranging between 2 and 6 glucopyranose units (termed nigerose, nigerotriose, etc.). Heating the assay reactions at 75 °C for 2 min was the minimum condition required to denature the enzyme and stop the catalysed reaction (Supplementary Fig. 2). Higher temperatures or prolonged incubation led to degradation of nigerooligosaccharides and, therefore, this condition was adopted as the standard methodology. The pH dependence was determined using 100 μM nigeropentaose as a substrate and revealed *An*GH71B as having >75% activity across the pH spectrum assayed (pH 4.5–8.5) while *An*GH71C was most active at pH 4.5 and had <50% activity at higher pH values, and all further assayed were performed at this pH for both enzymes (Supplementary Fig. 3). Neither of the two enzymes had any activity on nigerose and both had only minimal activity on nigerotriose. For the longer oligosaccharides the activity of *An*GH71B and *An*GH71C differed, both with regards to substrate preference and product profiles. *An*GH71B had the highest specific activity on nigerotetraose and slightly lower on nigeropentaose and nigerohexaose (Table 1). With nigerotetraose, the product profiles of both enzymes were the same, showing release mainly of glucose and nigerotriose and a minimal amount of nigerose. The products produced by *An*GH71B with nigeropentaose and nigerohexaose were predominantly glucose and nigerotriose with a smaller amount of nigerose which suggest a processive and *exo*-type of action on these substrates which cleaves off glucose units from one end of the oligosaccharide until the trisaccharide length is reached, which itself is subsequently hydrolysed to glucose and nigerose but at a much slower rate (Fig. 2f, g). In contrast, *An*GH71C preferred longer substrates and had the highest specific activity with nigerohexaose (Table 1), and in the reactions with nigeropentaose or nigerohexaose released a mixture of oligosaccharide products as well as glucose (Fig. 2i, j). In the reaction with nigerohexaose, the main products were nigerose, nigerotriose, and nigerotetraose with a smaller amount of glucose, which would be indicative of *An*GH71C preferentially hydrolysing longer nigerooligosaccharides in an *endo*-type fashion. The observation of differing potential modes of action, i.e. *endo*- versus *exo*-, has previously been observed in GH71[26–28] and in some other GH families, such as GH5[29], although to the best of our knowledge it is not a common occurrence within a GH family.

### *An*GH71B and –C are inverting enzymes with different specificities at the reducing end

Previous studies of *An*GH71C using carboxymethylated α-1,3-glucan as substrate revealed the enzyme to act in an inverting manner, producing the β-anomer of glucose[26]. Given the apparent preference for oligosaccharides for both *An*GH71B and –C, we sought to verify the proposed mechanism with the minimal length substrate nigerotetraose. Since both enzymes convert nigerotetraose to glucose and nigerotriose, using this substrate

would also ascertain information as to the preferred binding site(s) of the substrate, i.e. if it cleaves terminal moieties from the reducing or non-reducing ends of the oligosaccharide. When either GH71 enzyme was incubated with nigerotetraose, the products analysed by NMR spectroscopy showed a strong [1]H signal corresponding to β-nigerotriose and a racemic mixture of glucose anomers (Fig. 3a). The completed *An*GH71B reaction was tracked over 7 h where the mutarotation of the produced β-nigerotriose to the α-anomer was observed (Fig. 3b). Collectively, the results indicate that the enzymes preferentially cleave the glucose moiety at the reducing end of the tetrasaccharide by an inverting mechanism. To gain further insights into the enzymes' subsite preferences we evaluated the enzymatic products produced when utilizing chemically reduced versions of nigeropentaose and nigerohexaose, generated by sodium borohydride reduction (Fig. 3c–h). The products from both *An*GH71B and *An*GH71C with reduced nigeropentaose were similar, both producing nigerotriose and reduced nigerose. However, the enzymes differed in their profiles when using reduced nigerohexaose as a substrate, where *An*GH71B simultaneously produced similar amounts of glucose, nigerotriose and reduced nigerose, while *An*GH71C produced a mixture of nigerotriose, nigerotetraose, reduced nigerose and reduced nigerotriose. These observations support the proposed *endo*-lytic cleavage mode of *An*GH71C. For *An*GH71B, it was hypothesized that if the enzyme is *exo*-acting then reduced glucose would be produced and not the reduced nigerose as was observed. However, the lack of reduced nigerotriose being produced in the assays with reduced nigerohexaose supports the *exo*-acting proposal where the lack of reduced glucose is likely result from the enzyme's inability to bind the reduced unit in a conducive fashion in the +1 site. The activity of the enzymes towards the reduced oligosaccharides was lower than that of the native substrates, though the activity with *An*GH71C was only minimally affected indicating a lower specificity for the terminal reducing sugar configuration (Table 1).

### *An*GH71B and -C target α-1,3-glucan linkages embedded within the cell wall matrix

Recent advancements in the visualization of cell wall linkages by solid state NMR spectroscopy techniques has illuminated the presence of α-1,3-glucan linkages in the cell wall of *Aspergillus* species[2,30,31]. The glucan is found throughout the cell wall matrix with a greater quantity in rigid regions closer to the membrane but can additionally be found in the more mobile portions at the wall periphery[2]. Using solid state NMR spectroscopy, we explored if the two *A. nidulans* GH71 enzymes could cleave the α-1,3-glucan linkages found in intact cells, but we did not observe any significant changes in the α-1,3-glucan signals when the enzymes were externally applied (Supplementary Fig. 4a, b). The lack of cleavage could be a result of the enzymes being unable to penetrate the cell wall sufficiently to attack the target polysaccharides or that other CAZymes are needed to act in conjunction with the GH71 enzymes for deconstruction of the outer cell wall polysaccharides. However, when the fungal cells were disrupted by ball milling, exposing all cell wall components, and then treated with *An*GH71C, a mixture of glucose, nigerose, and nigerotriose was observed to be released via product analysis with HPAEC (Supplementary Fig. 4c). Collectively, the results indicate that these GH71 members can indeed cleave the α-1,3-glucan present in the *Aspergillus* cell wall but are unable to when solely applied externally to intact cells.

### *An*GH71C is a (β/α)₈ barrel intimately packed with a C-terminal β-sandwich

As no protein structural information is hitherto present for GH71 and to establish a greater fundamental understanding of the family, we pursued the structural determination of *An*GH71C by macromolecular crystallography. The structure was solved to 1.33 Å resolution by molecular replacement using a model of the protein generated by AlphaFold2[32,33] and all residues of the protein could be modelled into the electron density map, except for 9 and 12 N-terminal residues in chains A and B, respectively, and the N-terminal His₆ purification tag. Crystals of the apo protein and glucose-soaked crystals were of the C121 space group with similar unit cell dimensions and

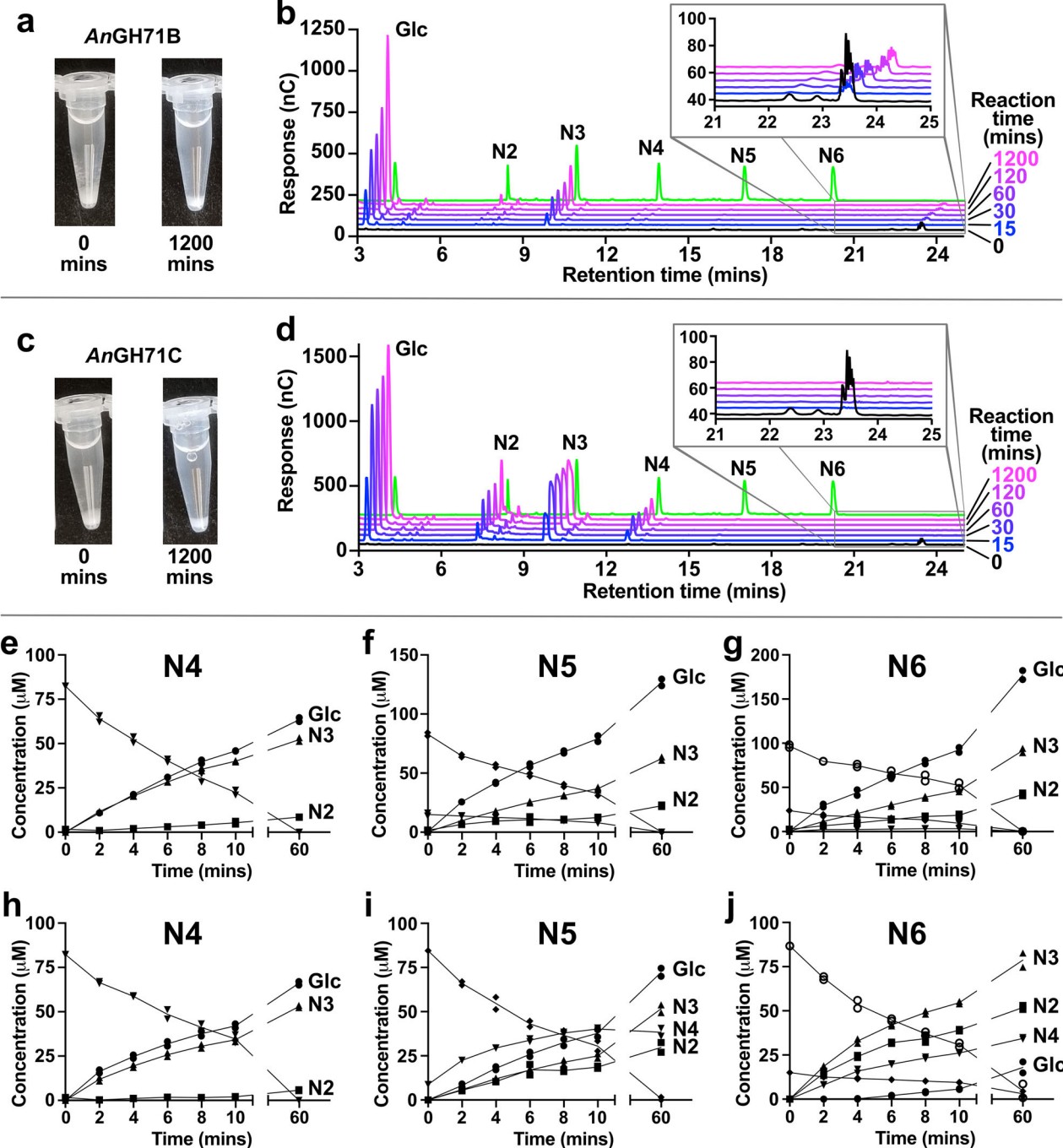

**Fig. 2 | Reactions with *An*GH71 enzymes indicate targeting of soluble oligo-saccharides.** Images of assays with 1% α-1,3-glucan before and after the 1200 min incubation with either (**a**) *An*GH71B or (**c**) *An*GH71C revealing that minimal insoluble material is hydrolysed during the enzyme incubations. Products produced from incubation of 0.5 nmol of either *An*GH71B (**b**) or *An*GH71C (**d**) with 1% α-1,3-glucan, shown alongside a run containing 100 μM of each nigerooligosaccharide standard (green), detected by HPAEC-PAD. The enzymes differ in their product profiles, where incubation with *An*GH71B leads to consistent production of glucose (Glc) and to a smaller extent nigerotriose (N3), while the *An*GH71C reactions rapidly produces Glc, nigerose (N2), N3, and nigerotetraose (N4) over 1 h, where-after the rate slows with only minimal additional product release over an additional 19 h. Note that the amounts for some peaks shown in the plots exceed the linear range commonly used for quantification (up to 100 μM) but was still quantifiable at higher concentrations through non-linear regression (Supplementary Fig. 14). The insets show soluble large molecular weight oligo- or polysaccharides which are rapidly degraded, more so by *An*GH71C, with concomitant production of low molecular weight products. Reaction progress curves of *An*GH71B (**e–g**) and *An*GH71C (**h–j**) with 100 μM of the oligosaccharide substrates nigerotetraose (N4, **e**, **h**), nigeropentaose (N5, **f**, **i**), and nigerohexaose (N6, **g**, **j**). The substrates and products shown are glucose (Glc, ●), nigerose (N2, ■), nigerotriose (N3, ▲), nigerotetraose (N4 ▼), nigeropentaose (N5, ♦), and nigerohexaose (N6, ○). Nigeropentaose and nigerohexaose were not completely pure (>90%) and contained minor amounts of nigerotetraose and nigeropentaose, respectively. Data points are from duplicate assays.

## Table 1 | Specific activity of *An*GH71B and –C with 100 µM nigerooligosaccharides

| Enzyme | Nigerotriose | Nigerotetraose | Nigeropentaose | Nigerohexaose | Nigeropentaose-reduced | Nigerohexaose-reduced |
|---|---|---|---|---|---|---|
| *An*GH71B | 2.50 ± 0.19 | 391 ± 19 | 316 ± 20 | 274 ± 23 | 16.4 ± 1.6 | 64.1 ± 8.3 |
| *An*GH71C | 0.144 ± 0.019 | 125 ± 7.7 | 791 ± 110 | 943 ± 47 | 230 ± 17 | 1130 ± 110 |

Activity (µmol min$^{-1}$ mg$_{protein}$$^{-1}$) was calculated from linear regression, with associated standard errors, of the rate of oligosaccharide consumption from two independent reactions measured every 2 min over a 10 min period measured by HPAEC-PAD. Reduced oligosaccharides were generated by NaBH$_4$ reduction as described in the methods. Reported values means and standard deviations from duplicate assays ($n$ = 2).

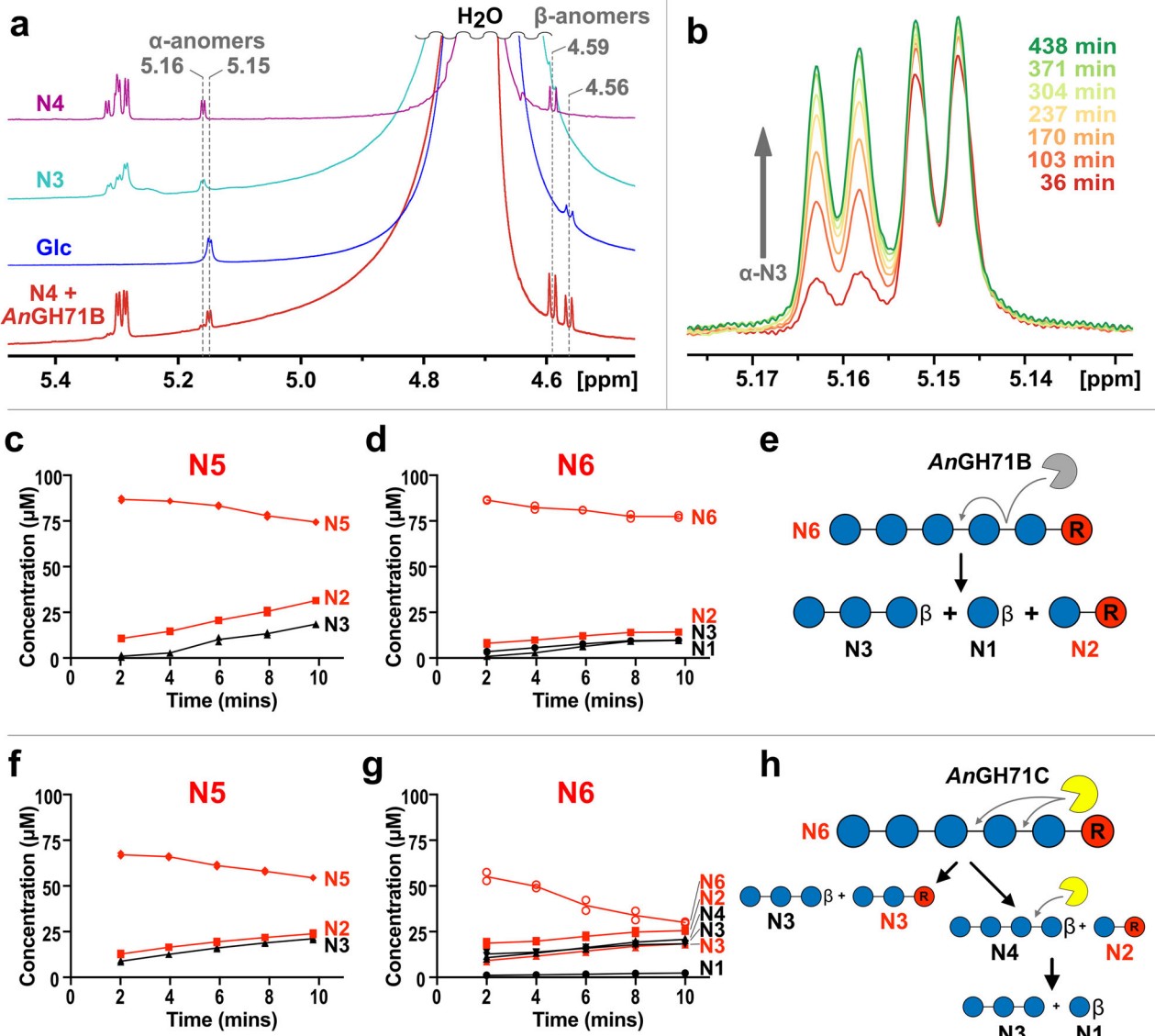

**Fig. 3 | Probing the GH71 mechanism. a** 1D $^1$H NMR spectra of nigerotetraose (N4), nigerotriose (N3), glucose (Glc), and nigerotetraose after 36 min incubation with 10 ng of *An*GH71B, a sufficient amount of enzyme and time to ensure at least ~90% turnover. Addition of enzyme leads to production of similar amounts of both anomeric forms of glucose (~5.15 and ~4.56 ppm for α and β, respectively) and a large amount of the β-anomer of nigerotriose (~5.16 and ~4.59 ppm for α and β, respectively), indicating cleavage of glucose occurs at the reducing end and proceeds through an inverting mechanism. **b** region of the $^1$H spectra of nigerotetraose treated with *An*GH71B over a 7-h incubation showing the build-up over time of the α-anomer (signal at 5.16 ppm) from mutarotation of the β-nigerotriose enzyme product. A slight build-up of the α-anomer of glucose (signal at 5.15 ppm) after the first

time point, 36 min, indicates the reaction is near completion over the initial period. Reactions of *An*GH71B (**c, d**) and *An*GH71C (**f, g**) with 100 µM of either NaBH$_4$ reduced nigeropentaose (**c, f**) or nigerohexaose (**d, g**) analysed by HPAEC-PAD. The substrates and products shown are glucose (N1, ●), nigerose (N2, ▪), nigerotriose (N3, ▲), nigerotetraose (N4, ▼), nigeropentaose (N5, ♦), and nigerohexaose (N6, ○) with the reduced version of the compounds coloured red. Data points are from duplicate assays. Illustration of the proposed cleavage reactions of reduced nigerohexaose performed by *An*GH71B (**e**) and *An*GH71C (**h**), with the formation of β-anomers at the reducing end resulting from the inverting mechanism.

**Table 2 | Table of crystallographic statistics**

| | *An*GH71C | *An*GH71C-glucose | *An*GH71C-nigerotetraose |
|---|---|---|---|
| Data Collection | | | |
| Date | September 18th, 2021 | September 18th, 2021 | November 26th, 2021 |
| Source | ID23-2 at ESRF | ID23-2 at ESRF | BioMAX at MAXIV |
| Wavelength (Å) | 0.87313 | 0.87313 | 0.97625 |
| Space group | C121 | C121 | P1 |
| Cell dimensions | | | |
| *a, b, c* (Å) | 137.19, 84.75, 103.32 | 135.02, 84.41, 102.22 | 80.27, 82.24, 87.07 |
| α, β, γ (°) | 90, 130.85, 90 | 90, 130.56, 90 | 103.52, 105.64, 117.59 |
| No. of measured reflections | 1,314,785 (29,218) | 1,174,865 (26,389) | 298,832 (14,770) |
| No. of unique reflections | 397,430 (11,634) | 353,567 (9796) | 82,756 (4138) |
| Resolution (Å) | 39.08 - 1.33 (1.35 - 1.33) | 42.09 - 1.37 (1.38 - 1.37) | 41.70 - 1.92 (1.94 - 1.92) |
| Ellipsoidal resolution limit (Å)[a] | – | – | 2.090 [0.750 a* − 0.658 b* + 0.064 c*] |
| | – | – | 2.646 [0.415 a* + 0.554 b* − 0.722 c*] |
| | – | – | 1.888 [−0.056 a* + 0.364 b* + 0.930 c*] |
| $R_{merge}$[b] | 0.087 (1.01) | 0.081 (1.01) | 0.118 (0.706) |
| $CC_{1/2}$[c] | 0.997 (0.314) | 0.997 (0.332) | 0.990 (0.524) |
| $\langle I/\sigma(I)\rangle$ | 7.62 (0.85) | 7.52 (0.81) | 6.4 (1.6) |
| Completeness spherical (%) | 99.6 (89.3) | 98.81 (85.82) | 62.5 (12.1) |
| Completeness ellipsoidal (%) | – | | 92.1 (65.5) |
| Redundancy | 3.3 (2.5) | 3.3 (2.7) | 3.6 (3.6) |
| Refinement | | | |
| $R_{work}/R_{free}$ | 0.159/0.176 | 0.163/0.182 | 0.164/0.221 |
| Number of protein, ligands, and water atoms | 6597, 118, 1256 | 6416, 48, 1012 | 12,740, 323, 688 |
| Average B-factors for protein, ligands, and water | 14.9, 19.3, 28.1 | 17.5, 21.7, 29.2 | 27.5, 29.2, 30.0 |
| Bond length (Å) and angles (°) RMSD from ideal geometry[d] | 0.008, 0.99 | 0.006, 0.89 | 0.008, 1.01 |
| Ramachandran favoured, allowed, and outliers (%) | 97.1, 2.5, 0.4 | 97.5, 2.0, 0.5 | 96.7, 2.9, 0.4 |
| PDB accession | 9FNF | 9FNG | 9FNH |

Data in parentheses is for the highest resolution shell.

[a]Brackets represent the direction along the reciprocal lattice.

[b]$R_{merge} = \sum_{hkl}\sum_{i}|I_i(hkl) - \langle I(hkl)\rangle|/\sum_{hkl}\sum_{i}I_i(hkl)$, wherein $I_i(hkl)$ is the intensity of the *i*th measurement of reflection *hkl*, and $\langle I(hkl)\rangle$ is the mean value of $I_i(hkl)$ for all the *i* measurements.

[c]Datasets for *An*GH71C and *An*GH71C-Glucose were processed in XDS[70] and truncated with XSCALE to a resolution where the outermost shell had a $CC_{1/2} > 0.5$ which were subsequently processed into finer width shells giving rise to $CC_{1/2}$ values < 0.5 in the outermost shell.

[d]Root mean square deviations from ideal geometry values[82].

contained 2 molecules in the asymmetric unit, while the nigerotetraose soak was determined in P1 containing 4 molecules in the asymmetric unit (Table 2). The protein contains an N-terminal (β/α)$_8$ barrel, a short linker (9 residues), and a C-terminal β-sandwich comprised of 8 strands which packs along the side of the barrel (Fig. 4a).

The β-sandwich domain has an immunoglobulin-like fold with a topology most closely related to Fibronectin type III (FnIII) members but contains an additional C-terminal β-strand (βE) compared to canonical members (Supplementary Fig. 5). The βE strand interacts with βA forming a four stranded sheet packing into a 4 × 4 sandwich which, to the best of our knowledge, is the first observation of such a topology. FnIII-like domains can also be found in some microbial glycolytic enzymes and a DALI search of structurally related proteins identified the FnIII-like domain from the multidomain chitin-active Auxiliary Activities family 10 (AA10) proteins from *Vibrio cholerae* and *Vibrio campbellii* as being most closely structurally related (Cα RMSD 2.8 Å compared to PDB accessions 2XWX and 8GUL) although sharing only 8% sequence identity[34,35]. As described in a recent review on AA10[36], the precise function of the FnIII-like domain in these proteins, termed module X, is unknown but the domain in the protein from *V. cholerae* (VcLPMO10B, previously referred to as GbpA) has been shown to be important for binding to the bacterial cell surface and for intestinal colonization[34] while other studies have suggested that the homologous domain does not function as a CBM[37,38] and could instead function as spacers separating catalytic domains from other units, as suggested for other CAZymes[39]. Multidomain AA10 members have been shown to adopt monomeric, elongated structures in solution, where the C-terminal module X makes few if any interactions with the other domains[34,35]. Conversely, in *An*GH71C, one face of the sandwich packs along, and partially buries, helices 7 and 8 of the (β/α)$_8$ barrel with two long loops of the sandwich between βA-βB and βG-βH extending up the barrel (Fig. 4a). The interface in *An*GH71C buries 21% of the sandwiches' solvent accessible surface, as determined by analysis with PISA[40], which includes several hydrophobic and aromatic residues along both faces and is stabilized by 14 hydrogen bonds and a salt bridge. Attempts to express either *An*GH71B or -C without their C-terminal domain led to insoluble proteins. We were, however, able to recombinantly produce the C-terminal domain from *An*GH71C, but not from *An*GH71B. Pull-down assays on a range of water-insoluble polysaccharides, including α-1,3-glucan, alternan (α1,3/α1,6-glucan), nigeran (BioSynth), potato starch, Avicel, cellulose microgranules, and xylan from beech wood, did not show any significant binding (Supplementary Fig. 6). Protein models of the four other GH71 enzymes from *A. nidulans* suggest similar folds and packing of the β-sandwich domain with the catalytic

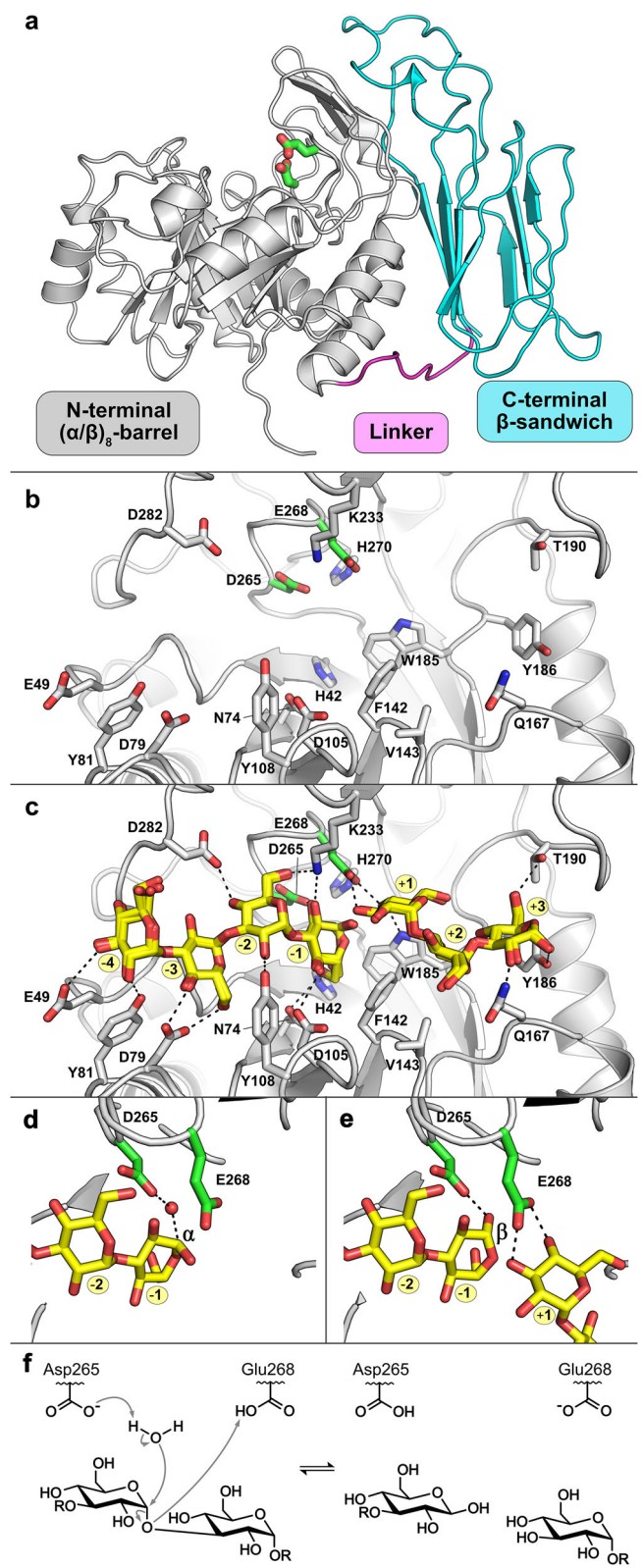

**Fig. 4 | Structure and catalytic mechanism of *An*GH71C. a** Overall structure of *An*GH71C with its N-terminal (β/α)$_8$-barrel, linker, and C-terminal β-sandwich domains shown in grey, magenta, and cyan, respectively. The conserved acidic residues in the active site in GH71 members are shown as green sticks protruding from cleft within the (β/α)$_8$-barrel. **b** The binding cleft in the absence of substrate and **c** with an overlay of the different nigerooligosaccharides observed in ligand structures from four protein molecules. Oligosaccharides are shown as yellow sticks with their proposed subsites highlighted in yellow. Hydrogen bonds (defined as electrostatic interactions between 2.8 and 3.2 Å) between the protein and ligands are shown as black dashes. Notably, the C-terminal β-sandwich domain (cyan) does not contribute to interactions within the binding site. The catalytic sites of chains A (**d**) and B (**e**) of the nigerotetraose soaked crystal revealing both α and β anomers at the reducing end of the soaked ligands. A water molecule (red sphere) is found positioned (≤3.2 Å) between the catalytic aspartate (Asp265) and the C1 of the glucose in the −1 subsite when the terminal sugars hydroxyl is in the α configuration, representative of the α-1,3-substrate, which is replaced by the hydroxyl in the β configuration, representative of the inverted product. **f** The proposed hydrolysis reaction mechanism where Asp265 acts as a general base to activate the water to act as the nucleophile leading to inversion about the anomeric carbon and Glu265 act as the general acid to protonate O3 leaving group.

The N-terminal (β/α)$_8$-barrel domain is most closely related to GH99 (Cα RMSD between 2.8 and 3.1 to GH99 members), a family mainly containing retaining *endo*-α-1,2-mannanases as characterized members[41] [,42]. The most distinct difference between the GH99 and GH71 (β/α)$_8$-barrels is that *An*GH71C lacks a region of ~50 residues found between β1 and α1 in the GH99 members, which forms a large loop wrapping one side of the substrate binding cleft (Supplementary Fig. 8). The loop between β8 and α8 in *An*GH71C occupies an equivalent position as the loop in the GH99 members and similarly builds up one side of the cleft, albeit to a lesser extent due to its shorter length which ultimately makes the cleft less deep. Complexes of *An*GH71C with glucose and nigerotriose/nigerotetraose were obtained by soaking crystals with ligand and clearly revealed the binding cleft within the N-terminal (β/α)$_8$-barrel domain (Fig. 4b, c). The two protein molecules of the glucose-soaked crystals each contain two glucose molecules located in the same position in the binding cleft, while each of the four protein molecules of the nigerotetraose-soaked crystals contain distinct combinations of nigerotriose and nigerotetraose occupying different subsites within the cleft (Supplementary Fig. 9). When overlaid and considered together, the annotation of the −4 to +3 subsites can be observed separated by two acidic residues (Asp265 and Glu268) found at the beginning of the loop between β8 and α8 of the (β/α)$_8$-barrel domain (Fig. 4c). An additional nigerotriose molecule was also observed on the surface of the cleft in chain B extending from the +3 subsite and could be bound in +4, +5, and +6 subsites though the trisaccharide makes few direct interactions with the protein and its position on the surface of the cleft makes the subsite annotations questionable and could result from soaking ligand in high concentrations (Supplementary Fig. 9d).

The two acidic residues are conserved amongst GH71 and GH99 members (Supplementary Fig. 10) where in the latter family they are found as a ExxE motif [41]. In silico simulations of nigerohexaose in water revealed preferred low energy φ and ψ configurations within the oligosaccharide linkages, similarly to what was observed in the ligands bound in the protein structures. This suggests that the enzyme may not need to significantly manipulate the oligosaccharide configuration to support binding in the subsites (Supplementary Fig. 11). However, bond angles of a heptasaccharide, generated by extrapolation of the observed ligands over the +1 to −1 sites, reveals a high energy configuration at this site which likely primes the oligosaccharide for cleavage. Only small protein structural changes are observed upon ligand binding in *An*GH71C with the most significant being the packing of the loop between β6 and α6 over the substrate along the +1 to +3 sites leading to interactions with Gly193 and Asn194 (Supplementary Fig. 12).

domain (Supplementary Fig. 7a), and although sequence identity is low amongst their β-sandwich domains (from 25 to 50% identity), sequence similarity is most prominent along the face of the sandwich packing against the (β/α)$_8$ barrel (Supplementary Fig. 7b, c). Taken together, the observations collectively point to an intimate and stable interaction between the (β/α)$_8$-barrel and β-sandwich domain, unlikely to dissociate in solution and maintained throughout GH71.

**Table 3 | Specific activity of *An*GH71 enzyme variants with 100 µM nigeropentaose**

| Enzyme | Specific activity (µmol min$^{-1}$ mg$_{protein}^{-1}$) | Fold reduction in activity from wild type |
|---|---|---|
| *An*GH71B D271A | 1.51 ± 0.12 | 210 |
| *An*GH71B E274A | 0.198 ± 0.032 | 1596 |
| *An*GH71C D265A | 0.129 ± 0.020 | 6142 |
| *An*GH71C E268A | 0.0460 ± 0.010 | 15,427 |

Activity was calculated from linear regression of the rate of oligosaccharide consumption as described in Table 1. Reported values means and standard deviations from duplicate assays (*n* = 2).

## Proposal for a classical inverting mechanism in GH71

The position of the acidic residues relative to the bound ligands and their conservation to GH99 members strongly suggested them to act as the catalytic residues for GH71 members. Substitution of either acidic residue in either *An*GH71B or –C led to a drastic loss of activity relative to the wild type (200–15,000-fold; Table 3) with reductions similar to that of the substitution of catalytic residues in some other GHs[43–45], and supports the roles for the conserved acidic residues in the mechanism of the enzymes. Within different chains of the oligosaccharide-soaked crystal, both α- and β-anomers of nigerotetraose were observed occupying the -4 to -1 subsites. When the α-anomer is present, resembling the substrate configuration, a water molecule is positioned close to Asp265, 3.2 Å from the C1 atom of the sugar at the reducing end (Fig. 4d). This position is taken over by the hydroxyl of the β-anomer, the configuration of the inverted product (Fig. 4e). Intensive studies of GH99 enzymes have supported the roles of the two acidic residues in retaining *endo*-α-1,2-mannanases to proceed through a non-classical mechanism via a 1,2-anhydro sugar intermediate[41,46–48]. In contrast to GH99, the ligand complexes with nearby water and kinetic results of the substitution variants of *An*GH71C however support a classical inverting glycoside hydrolase catalytic mechanism for α-1,3-glucan and/or niger-ooligosaccharide cleavage in GH71 whereby the conserved aspartate (Asp256 in *An*GH71C) acts as the general base to activate the bound water for nucleophilic attack with the conserved glutamate (Glu268 in *An*GH71C) acting as the general acid to protonate the leaving group (Fig. 4f).

## Discussion

α-1,3-glucans are present throughout microbial ecosystems, found in both fungal cell walls and in bacterial biofilms. However, our understanding of which enzymes are utilized for the degradation of such linkages and how the catalysis takes place is only partly studied. Degradative activity is thus far limited to GH87 and GH71, with the former family found almost exclusively in bacteria and the latter found in both bacteria and fungi. Aside from few exceptions[49,50], the enzymes from both families contain predicted signal peptides and have been shown or are expected to be extracellular[9]. Recent investigations into some GH87 members have elucidated the first protein structures and mechanism of the family[51–53]. However, similar detailed knowledge of GH71 has been lacking and our present exploration of the phylogeny, biochemistry, and structural biology of GH71 using *A. nidulans* enzymes as a model provides greater insights into this poorly explored family.

Although the here studied *An*GH71 enzymes were active on synthesized α-1,3-glucan polysaccharides, their activities were considerably more pronounced with nigerooligosaccharides, indicating that shorter chains that are possibly semi-soluble and thus exposed may be the target of the enzymes in vivo. The enzymes also had differences in their chain length preferences and product profiles which likely indicates similar but distinct usages of the enzymes by the cell. α-1,3-glucan has been found throughout the cell wall of *Aspergillus* species, found in both the mobile outer layer and the rigid inner portions, in addition to being found in both alkali-soluble and insoluble fractions[2,31]. Thus, a diverse set of α-1,3-glucanases are likely required to degrade, or remodel, this glycan depending on its cell wall context. Our

observation that the two *An*GH71 enzymes tested could not degrade α-1,3-glucan from intact cells when externally applied, but were able to on fractionated cell walls, is in contrast to recent studies of the GH87 α-1,3-glucanase from *Flavobacterium* sp. EK14 which could degrade α-1,3-glucan at the surface of the *A. oryzae* cell wall[25]. At this point, however, we cannot rule out that *An*GH71B and –C are able to degrade surface exposed α-1,3-glucan as its abundance and distribution throughout the fungal cell wall can be greatly influenced by morphotype, life cycle, and nutrient availability[49,54], and our experimental setup only probed one defined condition.

The potential targets of fungal GH71 enzymes are varied, with some enzymes found in secretomes, such as in *A. niger*[55], suggesting degradation of extracellular material, possibly for nutrient acquisition. In a related manner, the characterized α-1,3-glucanases from the mycoparasite *Trichoderma harzianum* has been suggested to be involved in predation[26]. Other fungal enzymes however, such as *An*GH71C (also referred to as MutA) and the two GH71 enzymes from *Schizosaccharomyces pombe* (Agn1p and Agn2p), localize to the cell wall and their expression is correlated with changes in life cycle which indicate cell wall remodelling functions for the enzymes[20,49]. Several fungal species contain multiple GH71 gene copies which are spread across the fungal branches of the phylogenetic tree, suggesting that the diversity in specificity observed in the two here studied enzymes may reflect a diversity in specificity across fungal enzymes throughout the family. α-1,3-glucan has an important role for fungal cell size[55–57], life cycle[58], and virulence for some pathogenic fungi, such as *A. fumigatus*[8], and how the repertoire of diverse GH71 enzymes within a fungus contribute to these modes requires further investigation and may advance both fundamental biological understanding and new antifungal strategies.

We elucidated the first protein structure of a GH71 member revealing a two-domain protein. The β-sandwich domain appears to be conserved throughout the family and is a distinguishing feature relative to homologous (β/α)$_8$-barrel hydrolases, such as GH99 members. However, the role of the domain in GH71 proteins remains elusive, as none of its residues are in close enough proximity to the substrate binding cleft to suggest a contribution during binding or catalysis. Additionally, its lack of observable polysaccharide binding ability, tight packing with the (β/α)$_8$-barrel domain, and our inability to recombinantly produce proteins in its absence point to a role in protein stabilization. Alternatively, the long loops protruding from the sandwich might be involved in substrate binding, as they pack against a substrate-interacting loop between β8 and α8 of the (β/α)$_8$-barrel, and an equivalent loop is also present in GH99 members. A conserved lysine residue (Lys 233 in *An*GH71C) is positioned above the catalytic dyad, and similar positively charged residues at the equivalent position are found in GH99 enzymes. Additional experimentally determined GH71 protein structures and a thorough investigation into the roles of each domain's components would however be needed to pinpoint the role of these β-sandwich domains.

In addition to the overall (β/α)$_8$-barrel structure, the active site organization of *An*GH71C is closely related to GH99 members with both comprising a tubular cleft housing the catalytic dyad motif D/E-x-x-E in a conserved position despite both proceeding through different reaction mechanisms[48]. Although the presence and position of the dyad is conserved amongst the two families, the identity of the first acidic residue in the dyad differs with aspartate found almost exclusively in GH71 and glutamate found almost exclusively in GH99. In ligand complex structures of the *Bx*GH99 endo-α-1,2-mannosidase from *Bacteroides xylanisolvens*, the substrate and 1,2-anhydro intermediate bind tightly to the protein surface and exclude solvent along the interacting face near the catalytic site[48]. *An*GH71C has an active site water molecule, proposed here to act as the catalytic nucleophile, positioned by the dyad's aspartate in a small gap along the protein surface and in line with the C1 of α-anomer products. This small gap, not present in ligand complex structures of GH99 members, principally results from the shorter aspartate side chain and, additionally, a change in the side chain and position of an aromatic residue (Trp185 in *An*GH71C) found adjacent to the first acidic residue of the dyad (Supplementary Fig. 13). The equivalent aromatic residue is a tyrosine residue in GH99

members (Tyr252 in *Bx*GH99) found on a loop shifted 2 Å towards the first glutamate of the dyad where its hydroxyl would lead to steric clashes with the active site water molecule found in *An*GH71C. Thus, although sharing a similar overall fold, active site architecture, catalytic residues, substrates, and substrate positioning, seemingly small changes around the catalytic core of GH71 and GH99 proteins lead to distinct differences in catalytic mechanism and present an interesting case for studying evolutionary enzymology.

GH71 and GH87 both contain enzymes capable of hydrolysing α-1,3-glucan linkages and both families utilize inverting mechanisms to catalyse the hydrolysis[51]. However, the two families are structural distinct from each other, with the catalytic unit of GH87 composed of a long right-handed β-helix[51–53]. GH87 enzymes contain a long binding site cleft positioned along the surface of one of the β-sheets, built-up by loops extending from the β-strands, where binding sites ranging from +4 to -4 have been observed but could extend further along the β-sheet[53]. GH87 members share structural similarity to GH28 and GH49, which all contain a similar right-handed β-helix and house the catalytic site comprised of three acidic residues in the centre of one of the β-sheets[59]. While the exact mechanistic details of GH87 enzymes are yet to be explicitly defined, biochemical analyses and similarity to their closely related families indicates two residues of the catalytic triad are expected to act as a general acid/base pair whereby the base abstracts a proton from a nucleophilic water to initiate the hydrolysis[51]. Thus, while containing distinct overall structures, both GH71 and GH87 contain relatively large binding site clefts and seem to utilize a similar general acid/base mechanism to facilitate the inverting hydrolysis of their target substrates.

In conclusion, this works presents novel insights into the atomic structure and catalytic mechanisms of GH71 enzymes, which have important roles in nature to deconstruct or remodel both fungal cell walls and bacterial biofilms such as dental plaque. The work lays a foundation for further studies of the diversity in substrate specificities and activities within the family, and may lead to new antifungal or oral health-related treatments. Notably, we have thus far only explored a small fraction of the diversity that exists within GH71, where the activity of the family's bacterial enzymes, representing approximately half of its known biodiversity, are yet to be investigated.

## Materials and methods
### Bioinformatics
The phylogenetic tree was created with all GH71 sequences the from CAZy database (July 2023) and sequences were trimmed relative to the catalytic domain of *An*GH71C. From the initial set of 543 sequences, identical sequences and fragments were removed, and the remaining sequences (355) were aligned using Clustal Omega[60]. The full-length protein sequences were initially annotated using the HMMER integration in dbCAN3[61] with an E-value $< 1e^{-15}$ and coverage $> 0.35$. Sequences containing large unannotated regions were assessed by structure predictions using AlphaFold2[33] to identify structural domains that were investigated with FoldSeek[62] and annotated where possible. The identity, placement, and length of the domains were visualized together with the phylogenetic tree using the protein domain dataset in iTOL[63].

### Chemicals
α-1,3-glucan and alternan (alternating α-1,3/α-1,6-linked glucan) polysaccharides were prepared enzymatically similar to methods previously described[64,65]. Briefly, genes for the glucosyltransferases GtfJ and GtfL from *Streptococcus salivarius* (ATCC 25975) were amplified from genomic DNA, using primers listed in Supplementary Table 1, cloned into pET28a, and the enzymes produced and purified as for *An*GH71C described in more detail below. Both polysaccharides were produced in 1 L volumes containing 10 mM sodium phosphate pH 6, 1 M sucrose, and 10 mg of either GtfJ, for α-1,3-glucan, or GtfL, for alternan. Reactions were left for 5 days at room temperature and gently mixed once a day. The polysaccharides were collected by either centrifugation, for α-1,3-glucan, or filtration using a 0.2-μm filter, for alternan, and were washed with water until no sucrose or fructose could be detected in the wash by an enzymatic assay kit (Megazyme,

K-SUFRG, detection limit of ~8 μM for each monosaccharide). Centrifugation was employed for α-1,3-glucan isolation as the material routinely obstructed the filter membrane. The materials were dried by lyophilization and stored at −20 °C until use. Nigerooligosaccharides (α-1,3-glucooligosaccharides) ranging in length from 2 (nigerose) to 6 (nigerohexaose) were obtained from Megazyme (catalogue numbers: O-NGR, O-NGR3, O-NGR4, O-NGR5, O-NGR6) and all other chemicals were from either Sigma Aldrich or Thermo Fisher unless otherwise stated.

### Molecular biology
GH71 genes were amplified from a cDNA preparation (RevertAid First Strand cDNA Synthesis Kit, Thermo Fisher Scientific) of *A. nidulans* FGSC A4 grown in YPD broth for 36 h. Screening of 10 constructs created for the *An*GH71-A, -D, and -E genes always led to constructs containing introns and the genes were instead obtained by gene synthesis and were codon optimized for expression in *E. coli* (Eurofins, Germany). PCR products of each *An*GH71B and –C, lacking their putative N-terminal signal sequence (determined by analysis with Signal P 5.0[66]), were ligated into a modified pET-28a vector, containing a TEV protease cleavage site instead of the native thrombin site, and transferred into *E. coli* BL21 (λDE3) for protein production. Enzyme variants were created by site-specific mutagenesis by the QuikChange method[67]. All constructs and gene mutations were verified by DNA sequencing. Primer sequences utilized for gene amplification and mutagenesis are provided in Supplementary Table 1.

### Protein production
Expression of the *An*GH71-A, -D, or -E constructs in *E. coli* (λDE3) led to insoluble protein production which could not be ameliorated by co-expression with chaperones (Takara Bio). Attempts to refold the proteins led to either insoluble aggregates or soluble protein which was inactive with nigerooligosaccharides suggesting improper folding. Further attempts to produce the proteins in *Pichia pastoris*, by integration of pPICZα-based vectors into strain X-33 were unsuccessful, possibly due to codon incompatibility from optimization for *E. coli*. For *An*GH71C, cells were grown in antibiotics-supplemented lysogeny broth (LB) at 37 °C and 200 rpm shaking, and at an $OD_{600} \sim 0.5$ expression was induced by addition of isopropyl β-D-1-thiogalactopyranoside (IPTG) to a final concentration of 0.2 mM and the cells incubated at 16 °C overnight. *An*GH71B was co-expressed with translation elongation factor (tig) from pTf16 (Clontech Laboratories) to yield sufficient soluble protein. For *An*GH71B, chaperone expression was induced at an $OD_{600} \sim 0.3$ by addition of 1 mg/mL L-arabinose, followed by IPTG induction at $OD_{600} \sim 0.5$. For both GH71 enzymes, cells were harvested by centrifugation (5000 × *g*, 10 min), followed by resuspension in 20 mM tris(hydroxymethyl)aminomethane (Tris) buffer pH 8 containing 250 mM NaCl, and disruption by sonication. Cell debris was removed by centrifugation (18,000 × *g*, 10 min), and proteins were purified using immobilized metal ion affinity chromatography on an ÄKTA system (GE healthcare) with 5 mL HisTrap™ Excel columns, using 50 mM Tris pH 8 with 250 mM NaCl as binding buffer and linear gradient of the same buffer containing 250 mM imidazole. The protein samples were concentrated by ultrafiltration (Amicon Ultra-15, Merck-Millipore), loaded onto a HiLoad Superdex 200 16/60 gel filtration column, and resolved with an isocratic gradient with the binding buffer. The protein samples were concentrated by ultrafiltration as before and stored at 4 °C. Sodium dodecyl sulfate polyacrylamide gel electrophoresis using Mini-PROTEAN TGX Stain-Free Gels (BIO-RAD) was used to verify protein purity and protein concentrations were determined using a Nanodrop 2000 Spectrophotometer (Thermo Fisher Scientific).

### Enzyme assays
Assays to measure specific activity were performed in 50 μL reactions containing 20 mM sodium acetate buffer pH 4.5, 100 μM nigerooligosaccharide, and *An*GH71B or *An*GH71C, with rates of cleavage determined based on reduction of substrate. Reactions were incubated at 25 °C with mixing at 750 rpm and stopped at defined time points by heat inactivation of the

enzymes at 75 °C with mixing at 1000 rpm for 2 min, and precipitated protein removed by centrifugation. Determination of enzyme dependence on pH was carried out with 100 μM nigeropentaose in a constant ionic strength three-component buffer containing 50 mM TRIS-HCl, 25 mM acetic acid, and 25 mM 2-(N-morpholino)ethanesulfonic acid (MES), covering a pH range of 4.5–8.5[68]. Screening of possible polysaccharide hydrolysing activity was completed in 1.5 mL reactions containing 20 mM sodium phosphate buffer pH 7.5, enzyme, and 1.33% w/v of either α-1,3-glucan, alternan, dextran, starch, ivory nut mannan, isolichenan, or α-mannan from *Saccharomyces cerevisiae* (BioSynth). The reactions were incubated at room temperature for 24 h with vertical rotation and stopped as before. Reduced nigerooligosaccharides (nigerotetraose, nigeropentaose, and nigerohexaose) used in kinetic assays for cleavage pattern analysis were generated by incubating 10 μL of 1 M NaBH$_4$ dissolved in 100 μM NaOH with 40 μL of 10 mM oligosaccharide for 1 h at room temperature as similarly described[69]. The reduction reaction was then quenched by the addition of 20 μL 1 M acetic acid and diluted with 10 μL 200 mM sodium acetate buffer and an additional 320 μL 20 mM sodium acetate buffer. All enzyme reactions were analysed by high-performance anion-exchange chromatography with pulsed amperometric detection (HPAEC-PAD) on a Dionex IC-5000+ (Thermo Fischer Scientific) equipped with a Dionex™ CarboPac™ PA200 IC Column (Thermo Fischer Scientific). The gradient method used is described in detail in Supplementary Table 2.

### NMR spectroscopy

Assays for the determination of the anomeric configuration of the enzymatic hydrolysis products of nigerotetraose were completed in 500 μL with 2 mM nigerotetraose in 25 mM sodium acetate (pH 4.5) prepared in D$_2$O. Reactions were initiated by the addition of 10 ng of *An*GH71B and 1D $^1$H spectra were collected at different points over a 7-h incubation at 25 °C. Spectra of nigerotetraose reaction were compared to spectra obtained for 2 mM glucose and 2 mM nigerotriose. The spectra were recorded at room temperature on an 800 MHz Bruker spectrometer equipped with a TXO cryoprobe. Solid state NMR experiments were performed on a 400 MHz Bruker Ascend 400DNP spectrometer equipped with a 3.2 mm MAS DNP probe. Isotopically labelled *A. nidulans* was prepared by culturing in 100 mL of a modified Czapek-Dox medium substituted with 10 g/L $^{13}$C glucose and 2 g/L $^{15}$N sodium nitrate as sole carbon and nitrogen sources (see Supplementary Table 3 for full recipe). Cells were grown for 1 week without shaking at 30 °C with the biomass harvested by filtration using a 0.2 μm nylon filter and washed with 20 mM sodium phosphate pH 7 prior to rotor packing. $^{13}$C direct polarization, cross-polarization and $^{13}$C,$^{13}$C-RFDR correlation experiments (10 kHz MAS, 1.5 ms mixing time) were performed on *A. nidulans* at room temperature before and after enzyme treatment with 1 μg of *An*GH71C. NMR signals belonging to cell wall components were assigned based on Kang et al.[2] and chemical shifts were externally referenced to the carbonyl signal of L-isoleucine.

### Macromolecular crystallography

Suitable crystallisation conditions for *An*GH71C were screened for using a Mosquito robot (SPT Labtech) with the JCSG+ screening kit (Molecular Dimensions, United Kingdom) in MRC sitting drop plates. The protein was dialysed into 50 mM Tris pH 8.0 containing 50 mM NaCl prior to screening. Screens were prepared with a reservoir volume of 40 μL, and protein was mixed with reservoir solution in a 1:1 ratio in 0.6 μL drops. Within 4 weeks, crystals of varying quality were observed in several of the conditions in the screen. Crystallisation conditions were optimised, and the final conditions used are listed in Supplementary Table 4. Crystals were mounted and flash frozen in liquid nitrogen in the absence of additional cryo-protectant. For ligand complexes, crystals were soaked in reservoir solution containing a saturating amount of ligand for 2 min prior to flash freezing in liquid nitrogen. X-ray diffraction data were collected at the ID23-2 beamline at the European Synchrotron Radiation Facility and at the BioMAX beamline of the MAX IV facility. An apo dataset of *An*GH71C diffracting to 1.33 Å was processed in XDS[70] and the structure solved by molecular replacement with

Phaser[71] in Phenix[72] using a model generated by AlphaFold2[32,33] as a search template. The dataset for the glucose complex was also processed by XDS[70], while the dataset for the nigerotetraose complex was anisotropic and after initial processing in XDS the data was elliptically truncated and corrected using the STARANISO server (http://staraniso.globalphasing.org)[73]. Structures for both ligand complex datasets were determined by molecular replacement with Phaser[71] in Phenix[72] using the apo protein as the template. For all structures, Coot[74] and Phenix Refine[75] were used in iterative cycles of real space and reciprocal space refinement. The data collection, processing, and refinement statistics for all datasets can be found in Table 2.

### Simulations of α-1,3-glucan configurations

Replica-exchange molecular dynamics simulations[76] of a single nigerohexaose oligomer in water was performed using GROMACS[77] at atmospheric pressure[78]. The GLYCAM06 force field[79] was used for the oligosaccharide and water was modelled with the SPC potential[80]. The temperature interval was 300 K to 375 K with 16 replicas spaced evenly every 5 K[81]. Conformational data (dihedral angles of the glycosidic bonds) was collected at 300 K. Total simulation time was 100 ns.

### Statistics and reproducibility

Specific activity values in Tables 1 and 3 are presented as mean ± standard deviation of the replicates from two independent assays, with the associated plots in Figs. 2 and 3 shown as individual values from the duplicate assays. The obtained replicates were in good agreement with each other. Linear regressions were used calculate rates using GraphPad PRISM software.

### Reporting summary

Further information on research design is available in the Nature Portfolio Reporting Summary linked to this article.

### Data availability

Coordinates and structure factors for *An*GH71C and the complexes with glucose and nigerotetraose have been deposited in the RCSB Protein Data Bank under accession codes 9FNF, 9FNG, and 9FNH, respectively. Raw and processed enzyme kinetic data from assays analysed by ion chromatography is provided in the Supplementary Data.

### Abbreviations

| | |
|---|---|
| GH | Glycoside Hydrolase |
| FnIII | Fibronectin type III |

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

## Acknowledgements

This work was primarily supported by funds awarded to J.L. by the Swedish Research Council Vetenskapsrådet (project 2020-03618), and partly by the Knut and Alice Wallenberg foundation via the Wallenberg Wood Science Center (WWSC). X-ray diffraction data were collected on the BioMAX beamline at MAX IV Laboratory (proposal 20200093) and beamline ID23-2 at the ESRF (proposal MX-2335). All NMR measurements were carried out at the Swedish NMR Centre in Gothenburg, Sweden.

## Author contributions

J.L. coordinated the project. S.M., T.W. and J.L. planned the experimental work. S.M. and G.B. solved the structure of *An*GH71C. H.K. and L.E. conducted the N.M.R. experiments. P.R and J.W. performed the computational modelling of α-1,3-glucan. S.M. and T.W. conducted the bioinformatic analyses and performed all other experimental tasks. S.M., T.W., H.K. and J.L. wrote the manuscript, with input from all other authors.

## Funding

## Competing interests

The authors declare no competing interests.
