## [Transparent Peer Review file · Communications Biology]

Structural and biochemical basis for activity of *Aspergillus nidulans* α -1,3-glucanases from glycoside hydrolase family 71

Corresponding Author: Dr Scott Mazurkewich

Version 0:

Reviewer comments:

Reviewer #1

(Remarks to the Author)

The manuscript submitted by Mazurkewich et al reports the characterization of GH71 alpha 1,3-glucanases from *Aspergillus nidulans*. CAZymes acting on fungal cell wall polysaccharides have been understudied and the GH71 family is no exception. Alpha 1,3-glucans play important roles in filamentous fungi and yeast with and therefore the study of this enzyme family is relevant both fundamentally and for applications in biotechnology. In this study, the authors focused on two fungal GH71s and performed in depth characterization to understand the structural requirements for enzyme activity and catalytic mechanism (inverting). Lots of interesting data are provided. The work is technically sound and the main conclusions are supported by the data. I have a few minor comments that may help the authors to improve the manuscript.

Introduction:

- The authors should start the introduction with the fungal cell wall context (line 54) as it is the main focus of the study and then mention the dental plaque biofilm context.
- line 64: the authors could mention that *C. neoformans* is the causative agent of meningitis in humans.
- line 74: are alpha-1,3 glucans found in barley straw? Surprising to see that GH71s have been used to saccharify lignocellulosic biomass? Please comment.
- line 87: What about the enzyme partially characterized (on mutan) in Bauer et al PMID: 16844780. Does it correspond to one of the 5 GH71 considered in this study?

Results:

- line 110: which CBMs? which additional catalytic modules? What is their specificity? Please add info in the main text.
- Figure 1 caption: provide the number of sequences used to build the tree
- line 120: mention in the results section the host and provide information about production yield and purification method. Did the authors tried other hosts (*P. pastoris*, *A. niger*)?
- line 150: I imagine that nigerose corresponds to DP2? Why is it not called nigerobiose? To avoid confusion, the authors should introduce this naming somewhere in the text for all oligos (DP2-DP6).
- line 187: the authors speculate that the enzyme could not reach the alpha 1,3 glucan but the lack of activity could also be due to the fact that other enzymes are required (other GH71s or other FCW active CAZymes)?
- line 227: The list of substrates tested should be mentioned in the main text. Was chitin tested?

Below some general comments that could be useful for the authors for future studies and discussion of the data (I am not asking additional experiments):

- The role of the C-term domain is intriguing. Do the authors think that the role of this domain is to maintain GH71 in contact with the fungal cell wall (as GPI anchors)? It would have been nice to perform microscopic (or proteomic) investigations to localize GH71s.
- Are GH71s secreted enzymes (it seems to be the case from the sequence and analysis with signalP but I did not find the info in the manuscript)? I think some transcriptomic and proteomic/secretomic data are available for *A. nidulans*. Did the authors look at gene regulation and secretion pattern?

Reviewer #2

(Remarks to the Author)

This study investigated two mutanases from *Aspergillus nidulans* in detail and provided new experimental information on protein structures, which the authors correlated with comprehensive analyses of the reaction products. This approach provides novel insights into the reaction mechanism of GH71 enzymes and will be useful for future studies in the area. The manuscript is well written, although I have highlighted some linguistic errors. I appreciate that the authors also report on failed enzyme expression in a detailed way. The material and methods are generally presented in such a way that they can be easily reproduced. Here are my suggestions for revision to improve the manuscript:

Abstract:

Line 32: grammar: "... could be exploited in biotechnological applications."

Line 37: the fact, that the enzymes are active on nigerooligosaccharides but not on large insoluble α 1,3-glucans is not surprising given the enzymes' accessibility to insoluble substrates. And please specify the term to "water-insoluble".

Line 39: are you referring to: fungal cell wall glycosidic linkages?

Line 42: I suggest to write "structure from GH71 family" instead of "structure from the family".

Introduction:

Line 51: expression: "...name mutan is derived".

Line 51: to the best of my knowledge, the polysaccharides are not initiating the dental plaque build-up, but promoting it.

Lines 58-60: please cite the original publications and indicate the years from which the data originate.

Line 65: plural: "have"

Line 75: please improve expression: "enhancement (or potentiation) of the effect of antimicrobial agents towards cariogenic bacteria".

Lines 80-82: please cite the original bioinformatics paper that grouped GH71 into two subfamilies.

Lines 82-83 & lines 113-115: this statement is not supported by the depicted results. Please label more corresponding branches in Figure 1 or provide literature.

Lines 84-88: please comment on your renaming of enzymes known in the literature. Is A through-E representing the sequence of the corresponding genes in the genome?

Results:

Lines 105-108: please provide more detailed information on the broader sequence analysis of the plant species as this maybe relevant for future studies exploring mutanases in plants. Which database and which bioinformatics tool did you use to find further plant sequences? And please provide protein accession numbers for the examples you give.

Line 126: why did you not test cellulose (β -glucan) and, if available, (partially) β -1,3-linked glucans in addition to β -mannan? In terms of substrate specificity, the investigation of β -linked glucans is as valuable as the testing of mannans. Given the structural similarity of the GH71 enzymes to GH99 enzymes, it would also be interesting to test the activity of GH71 against α mannan. Please provide data on enzyme activities on cellulose and, if accessible, on (partially) linked β -1,3-linked glucan and α -mannan.

Line 127: how did you determine the proportion of < 1%?

Lines 129 & Figure 2b,d: have you checked that the minimal increase in peak intensities is not due to reaching the limit of the linear PAD response (especially for the peak of nigerotriose)? Please provide chromatograms of samples with higher peak areas, e.g. of the oligosaccharide standard substances, or of lower concentrated hydrolysates to show that the minimal increase is not a matter of detector response.

Line 133 & Figure 2b,d: when does nigerohexaose (or your largest standard oligosaccharide) elute using your gradient? Could the substances eluting at ~23.5 min be (small) semi-soluble polysaccharides? Because they only elute with a high concentration of acetate.

Figures 2e-j: for a more intuitive understanding, I suggest adding the educt oligosaccharide as the graph title, e.g. "N4" in the middle of diagram e.

Figure 2 caption (Line 670): change sentence to "with concomitant production".

Line 710 (table 1): please add the method used to analyze the (reduced) oligosaccharides to the table caption (HPAEC-PAD). And please provide information on the standard errors (in the table caption or experimental section) for the determination: was it in triplicate?

Line 146: please check the reference to Fig. 2b-d.

Lines 160-161: please add that you are referring to signals in ^1H NMR spectra.

Lines 162-163: please move the conclusion that the enzyme preferentially releases β -N3 after the explanation in lines 163-164, as this information is crucial to the conclusion. And please add the observation (the signals of the α -anomeric protons increase with time) to lines 162-164.

Figures 3c,d,f,g: I suggest labelling the diagrams with the corresponding educt as proposed for Figures 2e-j and adding the analytical approach (HPAEC-PAD) in the Figure caption (line 690).

Line 678: please specify: " ^1H NMR spectra..."

Line 691: please specify: "Illustration of the proposed cleavage reactions..."

Lines 173-176: AnGH71B is not a nigerobiose releasing exo-acting enzyme because it releases high amounts of glucose (Figure 2)? Please discuss the aspect of processivity more in-depth in the text.

Line 190: please add information on the method (HPAEC vs. NMR spectroscopy above).

Line 715 (Table 2 caption): please add information in the analysis used (analyzed as given in Table 1?).

Lines 275, 276 & 283: Please check the reference to Figure 6.

Discussion:

Lines 290-293 and further down: given the fact that GH71 & GH87 share the same activity, I would appreciate it if the authors would compare the GH71 structure with known GH87 structures and point out the differences between the two. In addition, a brief comparison of the mechanism of both would be helpful.

Material and methods:

Line 369: please add how many sequences were finally aligned.

Line 388: please specify chain length of nigerooligosaccharides.

Line 394: please correct to "were synthesized".

Line 406: please comment on the codon incompatibility given that you used synthetic genes (where the codons can be optimized for *p. pastoris*).

Line 426: a pH of 4.5 seems low compared to the optima of other fungal mutanases. Why did you choose this pH?

Line 427: as for the pH a temperature of 25°C seems low compared to the optima of other fungal mutanases. Why did you choose this temperature?

Lines 428-429: did you proof that inactivation at 75°C for 2 min is complete? Usually, glucanhydrolases are inactivated at higher temperatures for longer durations.

Lines 433-437: for clarification, please add information on which oligosaccharides have been reduced and for what purpose.

Line 446: please add the temperature used.

Line 462: I suggest writing "...dialysed in 50 mM Tris buffer at pH 8.0..."

Supplementary:

Table 3 (caption): are the gradients linear?

Table 5: please give the concentrations of the ligands as you refer to them in the text (line 254).

Figure 4: in the PAGE it seems that the protein concentration varies, have you quantified the remaining protein in the supernatant after centrifugation? And please include a reference or short protocol (in the capture) showing the protocol used for PAGE & staining.

Reviewer #3

(Remarks to the Author)

The manuscript is a good characterisation and structural study of two Gh71 enzymes from *Asperigillus nidulans*. The authors contribute new knowledge to the GH71 family including identifying potential catalytic residues and a new structure. Overall the authors present a good body of data and are able to make interesting conclusions from it furthering our knowledge of this family of enzymes. Below are the minor concerns I have with the manuscript.

- The manuscript describes AnGH71B as exo active in the abstract, in the results section the authors are less confident stating line 146. "Potentially exo-type, of action on these substrates" The data in figure 2 doesn't seem to completely support an exo mode of action for the enzyme. Would you not expect an exo enzyme to show some activity on the disaccharide or trisaccharide even if it is exo-processive? This looks like the first time an exo mode of action has been described for this family with the other activities all being endo. Is it common to have both exo and endo action in the same GH family?

- The acidic residues that are proposed to be the catalytic residues do not kill the enzyme activity. Would you not expect when you mutate a catalytic amino acid that you lose all activity? Why do you not see this? Have you made a double mutant and does this retain its activity?

- Supp Fig 1. is incredibly hard to interpret, the modular architecture is almost impossible to make out. I think this needs re-designed, maybe you can highlight a few of the different architectures from across the phylogenetic tree.

Version 1:

Reviewer comments:

Reviewer #1

(Remarks to the Author)

I would like to thank the authors for addressing my comments and to congratulate them for this very nice and informative study. The manuscript will be very valuable for the community.

Reviewer #2

(Remarks to the Author)

The manuscript has been thoroughly revised to my complete satisfaction (all concerns cleared, text and figures improved), resulting in a publication that is both comprehensible and detailed. In my opinion, it should be published, as I believe it will benefit many others.

Reviewers' comments:

Reviewer #1 (Remarks to the Author):

The manuscript submitted by Mazurkewich et al reports the characterization of GH71 alpha 1,3-glucanases from *Aspergillus nidulans*. CAZymes acting on fungal cell wall polysaccharides have been understudied and the GH71 family is no exception. Alpha 1,3-glucans play important roles in filamentous fungi and yeast with and therefore the study of this enzyme family is relevant both fundamentally and for applications in biotechnology. In this study, the authors focused on two fungal GH71s and performed in depth characterization to understand the structural requirements for enzyme activity and catalytic mechanism (inverting). Lots of interesting data are provided. The work is technically sound and the main conclusions are supported by the data. I have a few minor comments that may help the authors to improve the manuscript.

Introduction:

The authors should start the introduction with the fungal cell wall context (line 54) as it is the main focus of the study and then mention the dental plaque biofilm context.

- This is a very good point, and we have now modified the abstract and introduction to frame the fungal cell wall context as the main focus.

line 64: the authors could mention that *C. neoformans* is the causative agent of meningitis in humans.

- This information has been included.

line 74: are alpha-1,3 glucans found in barley straw? Surprising to see that GH71s have been used to saccharify lignocellulosic biomass? Please comment.

- To the best of our knowledge, alpha-1,3 glucan is not reported to be present in barley straw, or in other cereals, and the report of the GH71 member in the rumen mixed enzyme extract providing benefit for the saccharification of barley straw was also surprising to us. Given their experimental setup, there are multiple possible reasons for why an improved saccharification effect was seen, but we do not believe it to be the GH71 enzymes directly hydrolyzing straw (i.e. β -linked glycans such as cellulose, xylan, or mixed-linkage glucan). There could simply be a correlation of GH71 enzymes co-migrating in the separation experiments they used with other enzymes with more relevant activities and showing up as a false positive, they could help prevent unproductive binding of other enzymes similar to how BSA has been shown to improve saccharification despite lacking enzymatic activity, or possibly by having a non-negligible side activity on other glucans such as starch which would be present in their experimental setup that included ruminal fluid from cows fed not only with barley straw, but also distillers grains. We feel that it would become confusing to the reader to go into a long speculation about this former observation, and have instead rephrased the sentence to:

“... and there is also a report linking improved barley straw saccharification by enzyme cocktails to presence of GH71 enzymes¹⁷, though the increase in released sugars could stem from other factors than direct straw hydrolysis, as α -1,3-glucan is not a known component of barley straw or other cereals.”

line 87: What about the enzyme partially characterized (on mutant) in Bauer et al PMID: 16844780. Does it correspond to one of the 5 GH71 considered in this study?

- We thank the reviewer for identifying this article to us. Yes, the mutanase (AN7349.2) from their studying corresponds AnGH71C in our study and we have included it as a citation here.
- Notably, they report on the activity present in the extract from the culture filtrate from a *Pichia* expression (ie. unpurified and not standardized to protein content) and it is thus not possible to be used for direct comparisons with our results.

Results:

line 110: which CBMs? which additional catalytic modules? What is their specificity? Please add info in the main text.

- We have expanded this portion of the text to include some specifics to what is being observed. Notably, that CBM24s are found, and the inaugural member of the family had been shown to bind α -1,3-glucan.

Figure 1 caption: provide the number of sequences used to build the tree

- We thank the reviewer for pointing out this omission. The caption for figure 1 and the associated portion of the methods has been updated to indicate that 355 sequences were used.

line 120: mention in the results section the host and provide information about production yield and purification method. Did the authors try other hosts (*P. pastoris*, *A. niger*)?

- We have updated this section to elaborate on the protein production yields and trials. We also refer the reader to the methods section where we further elaborate the attempts made with *AnGH71A*, -D, and -E.

line 150: I imagine that nigerose corresponds to DP2? Why is it not called nigerobiose? To avoid confusion, the authors should introduce this naming somewhere in the text for all oligos (DP2-DP6).

- Nigerose does indeed correspond to DP2. The utilization of the name nigerose instead of nigerobiose has historical roots and its utilization is akin to maltose and malto-oligosaccharides, where maltobiose is never used. While utilization of nigerobiose can be found in literature, nigerose is the common name for the disaccharide.
- We have expanded on the following line (expansion underlined) to improve the clarity of the naming of the oligosaccharides used.
 1. ...we characterized their specificity with commercially available nigerooligosaccharides, ranging between 2-6 glucopyranose units (termed nigerose, nigerotriose, etc.).

line 187: the authors speculate that the enzyme could not reach the alpha 1,3 glucan but the lack of activity could also be due to the fact that other enzymes are required (other GH71s or other FCW active CAZymes)?

- This is a very good point, and we thank the reviewer for bringing this to light.
- We have now edited this section to incorporate this point.

line 227: The list of substrates tested should be mentioned in the main text. Was chitin tested?

- The list of ligand compounds has now been included in the main text. We had not previously tried with chitin but have now done so and included it as a new panel in the supplemental figure. However, it too did not show any significant binding.

Below some general comments that could be useful for the authors for future studies and discussion of the data (I am not asking additional experiments):

The role of the C-term domain is intriguing. Do the authors think that the role of this domain is to maintain GH71 in contact with the fungal cell wall (as GPI anchors)? It would have been nice to perform microscopic (or proteomic) investigations to localize GH71s.

Are GH71s secreted enzymes (it seems to be the case from the sequence and analysis with signalP but I did not find the info in the manuscript)? I think some transcriptomic and proteomic/secretomic data are available for *A. nidulans*. Did the authors look at gene regulation and secretion pattern?

- The role of the C-term domain is indeed intriguing, and its purpose is still undetermined. Our results presented indicate that the domain packs tightly against the TIM barrel by residues and features seemingly conserved amongst the other GH71 members from *A. nidulans*. Furthermore, in AnGH71C and other predicted GH71 members, the C-term domain ends immediately or very shortly after the last β -strand, leaving little room for GPI associated mechanisms to facilitate connecting the linkage. Thus, while we can't as of yet rule out GPI-anchoring, the domains close association with the TIM barrel and short C-terminal beyond the domain make it unlikely to be utilized for this purpose.
- The localization of the proteins is an intriguing question and is something we are considering how best to pursue. In the method section we mention:
"PCR products of each AnGH71B and -C, lacking a putative N-terminal signal sequence, were ligated into a modified pET-28a vector..."
and in the discussion section we mention
"Aside from few exceptions^{39, 40}, the enzymes from both families (GH87 and GH71) contain predicted signal peptides and have been shown or are expected to be extracellular¹."
- Given the multiple copies of the GH71 genes in the *A. nidulans* genome, one could rationalize that they could be performing discrete functions or similar functions in different locations. The signal peptides indicate they are secreted out of the cytoplasm, but their fate is unclear at the moment. The lack of easily accessible α -1,3-glucan in nature suggests that they are not secreted to metabolize nutrient sources and thus points to a role in remodelling its own cellular environment. However, much work still needs to be completed to fully elucidate this, and we thank the reviewer for the suggestions on studies to pursue.

Reviewer #2 (Remarks to the Author):

This study investigated two mutanases from *Aspergillus nidulans* in detail and provided new experimental information on protein structures, which the authors correlated with comprehensive analyses of the reaction products. This approach provides novel insights into the reaction mechanism of GH71 enzymes and will be useful for future studies in the area. The manuscript is well written, although I have highlighted some linguistic errors. I appreciate that the authors also report on failed enzyme expression in a detailed way. The material and methods are generally presented in such a way that they can be easily reproduced. Here are my suggestions for revision to improve the manuscript:

Abstract:

Line 32: grammar: "... could be exploited in biotechnological applications."

- This grammatical error has been corrected.

Line 37: the fact, that the enzymes are active on nigerooligosaccharides but not on large insoluble α 1,3-glucans is not surprising given the enzymes' accessibility to insoluble substrates. And please specify the term to "water-insoluble".

- The terminology has been updated.

Line 39: are you referring to: fungal cell wall glycosidic linkages?

- Yes, and thank you for pointing out this omission.

Line 42: I suggest to write "structure from GH71 family" instead of "structure from the family".

- The term GH71 is an abbreviation for "glycoside hydrolase family 71" and so "GH71 family" would be redundant. However, we see the point the reviewer is making and have rewritten the passage for clarity as shown below.

"We present the first structure of a GH71 protein, AnGH71C, including structures with carbohydrate ligands bound in the active site. These structures revealed a conserved acidic dyad (DxxE), found to be crucial for activity, and active site water coordination consistent with a classical inverting glycoside hydrolase mechanism."

Introduction:

Line 51: expression: "...name mutan is derived".

- We have included this much improved phrasing.

Line 51: to the best of my knowledge, the polysaccharides are not initiating the dental plaque build-up, but promoting it.

- You are correct and the phrasing here has been corrected.

Lines 58-60: please cite the original publications and indicate the years from which the data originate.

- We have included the original publication as an additional citation and updated the text for this passage to as follows:

"Invasive fungal infections have high mortality rates, 20-95%, with Aspergillus fumigatus alone estimated in 2012 to be responsible for over 200,000 life-threatening infections annually world-wide^{5,6}."

Line 65: plural: "have"

- This grammatical error has been corrected.

Line 75: please improve expression: "enhancement (or potentiation) of the effect of antimicrobial agents towards cariogenic bacteria".

- This improvement has been incorporated.

Lines 80-82: please cite the original bioinformatics paper that grouped GH71 into two subfamilies.

- Upon further investigation we realize that we had misinterpreted the citation that we had provided. To the best of our knowledge, no one has extensively investigated the phylogenetic tree including entries from both bacteria and eukaryotes. We have now included a citation where the phylogeny of the fungal portion was examined and modified the text accordingly.

Lines 82-83 & lines 113-115: this statement is not supported by the depicted results. Please label more corresponding branches in Figure 1 or provide literature.

- We thank the reviewer for identifying this point. We have now included some additional entries from fungal species to help illustrate this point.

Lines 84-88: please comment on your renaming of enzymes known in the literature. Is A through-E representing the sequence of the corresponding genes in the genome?

- We thought it logical to have named the genes and gene products in numeric order by which they are described by locus tag accessions as described below.

AnGH71A	AN1604.2
AnGH71B	AN3790.2
AnGH71C	AN7349.2
AnGH71D	AN8252.2
AnGH71E	AN9042.2

- To clarify this to the reader we have included the accession codes when they are first described in the results section.

Results:

Lines 105-108: please provide more detailed information on the broader sequence analysis of the plant species as this maybe relevant for future studies exploring mutanases in plants. Which database and which bioinformatics tool did you use to find further plant sequences? And please provide protein accession numbers for the examples you give.

- We had utilized a BLASTp search using the entry from Sitka spruce as the query. We have updated the manuscript to include how the search was done and the associated Genbank accessions.

Line 126: why did you not test cellulose (β -glucan) and, if available, (partially) β -1,3-linked glucans in addition to β -mannan? In terms of substrate specificity, the investigation of β -linked glucans is as valuable as the testing of mannans. Given the structural similarity of the GH71 enzymes to GH99 enzymes, it would also be interesting to test the activity of GH71 against α mannan. Please provide data on enzyme activities on cellulose and, if accessible, on (partially) linked β -1,3-linked glucan and α -mannan.

- We had written that we had tried using α -mannan from *S. cerevisiae* but, like the other polysaccharides tested aside from mutan, did not observe any activity.
- We had previously not included β -glucans in our screen as from previous reports on members of the family did not suggest any activities on β -linked substrates, and dual activity on both α - and β -linked glycans in a single CAZy family is extremely unlikely. However, to be thorough, we have now tested AnGH71C on a range of β -linked polysaccharides, including barley β -glucan, yeast β -glucan, curdlan, konjac glucomannan, ivory nut mannan, micro crystalline cellulose, or carboxymethyl cellulose), but found not activity on any. We have updated the manuscript to include this investigation.

Line 127: how did you determine the proportion of < 1%?

- This was done by DNS assay and it is now described in the manuscript.

Lines 129 & Figure 2b,d: have you checked that the minimal increase in peak intensities is not due to reaching the limit of the linear PAD response (especially for the peak of nigerotriose)? Please provide chromatograms of samples with higher peak areas, e.g. of the oligosaccharide standard substances, or of lower concentrated hydrolysates to show that the minimal increase is not a matter of detector response.

- We thank the reviewer for pointing out that the peaks, particularly for that of nigerotriose in panel d of Figure 2, appear to be saturating the detector. It is true that some of the peaks shown exceed the linear range of detection. However, quantification is still possible via non-linear regression.
- To improve this figure and clarify these points, we have included nigerooligosaccharides at 100 μ M into panels b and d of Figure 2 and have supplied standard curves for the standards up to 2.5 mM for each (a range encompassing the peak areas observed) as Supplemental Figure 14.
- Notably, while non-linear regression may not be ideal for quantification, we believe in this case, where we state in more of a qualitative sense, that “...reactions rapidly produces Glc, nigerose (N2), N3, and nigerotetraose (N4) over 1 h, whereafter the rate slows with only minimal additional product release over an additional 19 h.” and that “... the amount of glucose and small oligosaccharides produced only minimally increased after 120 min (Fig. 2b and d).”, which we believe is an accurate depiction based on the results and quantification provided.

Line 133 & Figure 2b,d: when does nigerohexaose (or your largest standard oligosaccharide) elute using your gradient? Could the substances eluting at ~23.5 min be (small) semi-soluble polysaccharides? Because they only elute with a high concentration of acetate.

- We have now included a run containing 100 μ M of each nigerooligosaccharide standard into the figure panels. Nigerohexaose elutes at ~19.9 mins, before the substantial increase in NaAc is applied which subsequently enables elution of the large molecular weight compounds (oligo- or polysaccharides). We hope that the inclusion of the standards in the figure clarifies this point for the reader.

Figures 2e-j: for a more intuitive understanding, I suggest adding the educt oligosaccharide as the graph title, e.g. “N4” in the middle of diagram e.

- This is a great suggestion. We have incorporated oligosaccharide naming into the figure panels.

Figure 2 caption (Line 670): change sentence to “with concomitant production”.

- The phrasing has been updated.

Line 710 (table 1): please add the method used to analyze the (reduced) oligosaccharides to the table caption (HPAEC-PAD). And please provide information on the standard errors (in the table caption or experimental section) for the determination: was it in triplicate?

- The captions for Table 1 and 2 have been updated.

Line 146: please check the reference to Fig. 2b-d.

- This panels referred to have been corrected.

Lines 160-161: please add that you are referring to signals in ¹H NMR spectra.

- The reference to ¹H NMR spectra has been added.

Lines 162-163: please move the conclusion that the enzyme preferentially releases β-N3 after the explanation in lines 163-164, as this information is crucial to the conclusion. And please add the observation (the signals of the α-anomeric protons increase with time) to lines 162-164.

- This is a good suggestion to improve the clarity of the results to the reader and we have modified the text accordingly.

Figures 3c,d,f,g: I suggest labelling the diagrams with the corresponding educt as proposed for Figures 2e-j and adding the analytical approach (HPAEC-PAD) in the Figure caption (line 690).

- The diagrams in Figure 3c,d,f,g have been updated to include labelling as in the panels of Figure 2.
- The HPAEC-PAD methodology has been added to the caption.

Line 678: please specify: “¹H NMR spectra...”

- The text has been corrected to include “¹H NMR spectra”.

Line 691: please specify: “Illustration of the proposed cleavage reactions...”

- “proposed” is now included.

Lines 173-176: AnGH71B is not a nigerobiose releasing *exo*-acting enzyme because it releases high amounts of glucose (Figure 2)? Please discuss the aspect of processivity more in-depth in the text.

- We thank the reviewer for identifying an area where more clarity for the reader is required. We have also taken some time to reinterpret the results and have rewritten this section as described below:

“The products produced by *AnGH71B* with nigeropentaose and nigerohexaose were predominantly glucose and nigerotriose with a smaller amount of nigerose which suggest a processive and *exo*-type of action on these substrates which cleaves off glucose units from one end of the oligosaccharide until the trisaccharide length is reached, which itself is subsequently hydrolysed to glucose and nigerose but at a much slower rate (Fig. 2f & g). In contrast, *AnGH71C* preferred longer substrates and had the highest specific activity with nigerohexaose (Table 1), and in the reactions with nigeropentaose or nigerohexaose released a mixture of oligosaccharide products as well as glucose (Fig. 2i & j). In the reaction with nigerohexaose, the main products were nigerose, nigerotriose, and nigerotetraose with a smaller amount of glucose, which would be indicative of *AnGH71C* preferentially hydrolysing longer nigerooligosaccharides in an *endo*-type fashion. The observation of differing potential modes of action, ie. *endo*- versus *exo*-, has previously been observed in GH71^{26, 27, 28} and in some other GH families, such as GH5²⁹ although to the best of our knowledge it is not a common occurrence within a GH family. “

Line 190: please add information on the method (HPAEC vs. NMR spectroscopy above).

- The methodology used has now been included and the passage elaborated for clarity.

Line 715 (Table 2 caption): please add information in the analysis used (analyzed as given in Table 1?).

- The caption for Table 2 has been modified by to include the following

“Activity was calculated from linear regression of the rate of oligosaccharide consumption as described in Table 1.”

- Additionally, the caption for Table 1 has been modified to include that the measurements were taken by HPAEC.

Lines 275, 276 & 283: Please check the reference to Figure 6.

- This incorrect reference has been corrected to refer to the panels in figure 4.

Discussion:

Lines 290-293 and further down: given the fact that GH71 & GH87 share the same activity, I would appreciate it if the authors would compare the GH71 structure with known GH87 structures and point out the differences between the two. In addition, a brief comparison of the mechanism of both would be helpful.

- We thank the reviewer for pointing this out. We agree that a brief comparison of the two families is warranted and have now included a short paragraph comparing the two families as described below:

“GH71 and GH87 both contain enzymes capable of hydrolysing α -1,3-glucan linkages and both families utilize inverting mechanisms to catalyse the hydrolysis⁵¹. However, the two families are structural distinct from each other, with the catalytic unit of GH87 composed of a long right-handed β -helix^{51, 52, 53}. GH87 enzymes contain a long binding site cleft found along the surface of one of the β -sheets, built-up by loops extending from the β -strands, where binding sites ranging from +4 to -4 have been observed but could extend further along the long β -sheet⁵³. GH87 members share structural similarity to GH28 and GH49, which all contain a similar right-handed β -helix and house the catalytic site comprised of three acidic residues in the centre of one of the β -sheets⁵⁹. While the exact mechanistic details of GH87 enzymes are yet to be explicitly defined, biochemical analyses and similarity to their closely related families indicates two residues of the catalytic triad are expected to act as a general acid/base pair whereby the base abstracts a proton from a nucleophilic water to initiate the hydrolysis⁵¹. Thus, while containing distinct overall structures, both GH71 and GH87 contain relatively large binding site clefts and seem to utilize a similar general acid/base mechanism to facilitate the inverting hydrolysis of their target substrates.”

Material and methods:

Line 369: please add how many sequences were finally aligned.

- The passage has been modified to clarify the number of sequences analyzed as described below.

“From the initial set of 543 sequences, identical sequences and fragments were removed, and the remaining sequences (355) were aligned using Clustal Omega⁵⁰.”

Line 388: please specify chain length of nigerooligosaccharides.

- The chain length of oligosaccharides obtained has now been included.

Line 394: please correct to “were synthesized”.

- This grammatical error has been corrected.

Line 406: please comment on the codon incompatibility given that you used synthetic genes (were the codons can be optimized for *p. pastoris*).

- The following passages have been modified to clarify this point.
“Screening of 10 constructs created for the AnGH71-A, -D, and -E genes always led to constructs containing introns and the genes were instead obtained by gene synthesis and were codon optimized for expression in *E. coli* (Eurofins, Germany).”

“Further attempts to produce the proteins in *Pichia pastoris*, by integration of pPICZ α -based vectors into strain X-33 were unsuccessful, possibly due to codon incompatibility from optimization for *E. coli*.”

Line 426: a pH of 4.5 seems low compared to the optima of other fungal mutanases. Why did you choose this pH?

- We thank the reviewer for pointing out this omission. We have updated the manuscript to include a supplemental figure of the pH optimum completed (now Figure S2) and added associated sections in the results and methods as described below.
- Results
“The pH dependence was determined using 100 μ M nigeropentaose as a substrate and revealed AnGH71B as having >75% activity across the pH spectrum assayed (pH 4.5 to 8.5) while AnGH71C was most active at pH 4.5 and had < 50% activity at higher pH values, and all further assayed were performed at this pH for both enzymes (Supplementary Fig. 2).”
- Methods
“Determination of enzyme dependence on pH was carried out with 100 μ M nigeropentaose in a constant ionic strength three-component buffer containing 50 mM TRIS-HCl, 25 mM acetic acid, and 25 mM 2-(N-morpholino)ethanesulfonic acid (MES), covering a pH range of 4.5–8.5⁶¹.”

Line 427: as for the pH a temperature of 25°C seems low compared to the optima of other fungal mutanases. Why did you choose this temperature?

- As we were performing assays over a wide range of times, from short intervals for kinetics to over 24 to 48 hrs for product profiles and screening of potential substrates, we wanted to ensure protein stability throughout the long-time course to the best of our ability. We did not complete a temperature optimum for our enzymes. However, we can assume an increase activity with temperature up to a point where the rate of protein inactivation becomes substantial. Considering these points, we elected to simplify our approach and utilize a room temperature equivalent throughout our screening and kinetic investigations, and chose to leave the selection of temperature for the assays unmotivated in the manuscript.

Lines 428-429: did you proof that inactivation at 75°C for 2 min is complete? Usually, glucanhydrolases are inactivated at higher temperatures for longer durations.

- We had initially been boiling reactions at 95 deg but had found that at that high of a temperature, or with long exposures at increased temperatures, that nigerooligosaccharides would get degraded. We then performed a temperature screen to ascertain the minimum temperature required to stop the reaction and found 75 °C was sufficient. We have now included this investigation into the manuscript as the new

Supplemental Figure 2 and included text in the “Enzyme characterization reveals distinct specificities for nigerooligosaccharides” section of the results.

Lines 433-437: for clarification, please add information on which oligosaccharides have been reduced and for what purpose.

- Additional information is now provided.

Line 446: please add the temperature used.

- The temperature is now included.

Line 462: I suggest writing “...dialysed in 50 mM Tris buffer at pH 8.0...”

- We believe that “dialysed into” is the correct phrasing here as we moved the protein *into* a new solution, rather than moving the protein *in* a new solution, and thus have elected to keep the phrasing as is.

Supplementary:

Table 3 (caption): are the gradients linear?

- The caption has been updated and an image provided to improve clarity.

Table 5: please give the concentrations of the ligands as you refer to them in the text (line 254).

- The caption for supplementary table 5 has been updated to include that the solutions were saturating solution with ligand, as was already described in the methods (provided below).

“For ligand complexes, crystals were soaked in reservoir solution containing a saturating amount of ligand for 2 min prior to flash freezing in liquid nitrogen.”

Figure 4: in the PAGE it seems that the protein concentration varies, have you quantified the remaining protein in the supernatant after centrifugation? And please include a reference or short protocol (in the capture) showing the protocol used for PAGE & staining.

- We have also updated the caption to include more methodology related to the running of the SDS-PAGE.
- We have not analytically quantified the change in protein concentration. We utilized the pulldown assay as qualitative measure to assess if significant binding was occurring, which we did not see. Instead, the small variation in protein concentration is likely due to the insoluble polysaccharides trapping some protein when being spun down or some weak binding capability. Thus, from Figure S4, we concluded that:

Line 230 “Pull-down assays on a range of water-insoluble polysaccharides... did not show any significant binding (Supplementary Fig. 4).”

Reviewer #3 (Remarks to the Author):

The manuscript is a good characterisation and structural study of two Gh71 enzymes from *Aspergillus nidulans*. The authors contribute new knowledge to the GH71 family including identifying potential catalytic residues and a new structure. Overall the authors present a good body of data and are able to make interesting conclusions from it furthering our knowledge of this family of enzymes. Below are the minor concerns I have with the manuscript.

The manuscript describes AnGH71B as *exo* active in the abstract, in the results section the authors are less confident stating line 146. "Potentially *exo*-type, of action on these substrates" The data in figure 2 doesn't seem to completely support an *exo* mode of action for the enzyme. Would you not expect an *exo* enzyme to show some activity on the disaccharide or trisaccharide even if it is *exo*-processive? This looks like the first time an *exo* mode of action has been described for this family with the other activities all being *endo*. Is it common to have both *exo* and *endo* action in the same GH family?

- We thank the reviewer for raising this point about our description of the modes of substrate utilization, particularly around our interpretations and descriptions. Notably, our report is not the first time *exo*- and *endo*- activities are being reported for GH71 members. As described in the comments to Reviewer 2, we have rewritten these portions to improve the clarity and highlight that *exo*- and *endo*- activities have previously been reported in GH71.
- AnGH71B does have some activity on the trisaccharide, albeit it is >100-fold less than the tetrasaccharide (Table 1), and thus in the short timeframe with the amount of enzyme loading we would not expect to see much hydrolysis of the trisaccharide. Further, *exo*- activity does not necessarily mean that the oligosaccharides will be hydrolysed all the way to a monosaccharide at the non-reducing end and instead are usually arrested at disaccharides, such as with many *exo*-acting xylanases producing xylose and xylobiose with the later hydrolysed by xylosidases. Here, it is thus likely that α -glucosidases/nigerobiases would fulfil an equivalent role.

The acidic residues that are proposed to be the catalytic residues do not kill the enzyme activity. Would you not expect when you mutate a catalytic amino acid that you lose all activity? Why do you not see this? Have you made a double mutant and does this retain it's activity?

- Often substitution of catalytic residues in GHs does not completely kill activity. Instead, reductions of activity are usually seen in the 100 to 10,000-fold range which can be beyond the limit of detection for some systems. There are several possibilities for why activity is still present, most notably that binding of substrate occurs in a configuration which primes it for catalysis and for which bulk solvent could fulfil acid/base or nucleophilic roles.
- We have modified the text to include this point and provide citations to similar occurrences as described below:

Substitution of either acidic residue in either *AnGH71B* or –C led to a drastic loss of activity relative to the wild type (200-15,000-fold; Table 2) with reductions similar to that of the substitution of catalytic residues in some other GHs^{40, 41, 42}, and supports the roles for the conserved acidic residues in the mechanism of the enzymes.

Supp Fig 1. is incredibly hard to interpret, the modular architecture is almost impossible to make out. I think this needs re-designed, maybe you can highlight a few of the different architectures from across the phylogenetic tree.

- We agree that this figure is hard to interpret. We have modified it to more easily see the common architectures, increased the fonts on the proteins under investigation and labels, and have moved the figure to a larger size page. Additionally, we supply both pdf and vectors formats of the image which can be viewed with lossless resolution upon increasing magnification and can be utilized for searching of specific accessions.